# Spatiotemporal expression of regulatory kinases directs the transition from mitosis to cellular morphogenesis in *Drosophila*

Shuo Yang [1], Jennifer McAdow[1], Yingqiu Du[1], Jennifer Trigg[2], Paul H. Taghert[2] & Aaron N. Johnson [1]✉

Embryogenesis depends on a tightly regulated balance between mitosis, differentiation, and morphogenesis. Understanding how the embryo uses a relatively small number of proteins to transition between growth and morphogenesis is a central question of developmental biology, but the mechanisms controlling mitosis and differentiation are considered to be fundamentally distinct. Here we show the mitotic kinase Polo, which regulates all steps of mitosis in *Drosophila*, also directs cellular morphogenesis after cell cycle exit. In mitotic cells, the Aurora kinases activate Polo to control a cytoskeletal regulatory module that directs cytokinesis. We show that in the post-mitotic mesoderm, the control of Polo activity transitions from the Aurora kinases to the uncharacterized kinase Back Seat Driver (Bsd), where Bsd and Polo cooperate to regulate muscle morphogenesis. Polo and its effectors therefore direct mitosis and cellular morphogenesis, but the transition from growth to morphogenesis is determined by the spatiotemporal expression of upstream activating kinases.

[1] Department of Developmental Biology, Washington University School of Medicine, St. Louis, MO 63110, USA. [2] Department of Neuroscience, Washington University School of Medicine, St. Louis, MO 63110, USA. ✉email: anjohnson@wustl.edu

Rapid cell divisions are the hallmark of early embryonic development, but as development proceeds cells will exit the cell cycle to acquire specialized, functional morphologies. Tissue patterning through the differential expression of morphogens is a well-understood developmental process. On the other hand, kinases are generally considered to be stably expressed mediators of upstream activating proteins[1–5], and most kinases are ubiquitously expressed while the embryo is undergoing rapid cell divisions (Supplementary Fig. 1A and Supplementary Data 1). However, during tissue diversification and organogenesis, the expression of many kinases becomes enriched or depleted in specific tissues (Supplementary Fig. 1A and Supplementary Data 1). The spatiotemporal regulation of kinase expression could be a previously unrecognized and essential mechanism that drives the transition from mitosis to cellular morphogenesis.

Mitosis is controlled by two sets of kinases. Cyclin-dependent kinases regulate progress through the cell cycle, and mitotic kinases, including Aurora kinases and Polo-like kinases (Plks), direct mitotic entry, chromosome segregation, and cytokinesis[6–14]. Plk activity is dependent on two conserved protein domains. The C-terminal Polo-Box Domain (PBD) recognizes target substrates, and PBD docking enhances substrate phosphorylation by the kinase domain (KD)[11]. Intramolecular interactions between the PBD and the KD dictate the affinity of Polo for specific substrates. The PBD of *Drosophila* Polo kinase (Polo) binds the microtubule protein Map205 during interphase, which effectively sequesters Polo on microtubules[15]. During mitotic entry, activating phosphorylation of the KD by Aurora B (AurB) relieves PBD binding to Map205 and promotes PBD binding to pro-mitotic substrates and structures[16–18]. A similar regulatory mechanism has recently been proposed to modulate Polo activity during meiosis[19]. In addition, when Polo is inactive, intramolecular binding between the Polo KD and the PBD masks a nuclear localization signal (NLS). Activating phosphorylation exposes the NLS, allowing Polo to enter the nucleus prior to nuclear envelope breakdown[20]. Although Aurora-mediated Polo activation is an essential regulatory step during cell division, the role of Polo in post-mitotic tissues, if any, is unknown.

Cellular morphogenesis provides functionality for highly specialized cells, and morphogenesis generally initiates after cells have exited the cell cycle. Cellular guidance is a cytoskeleton-dependent morphogenetic process in which a post-mitotic cell generates long projections that interact with or connect to other cells. Axon guidance is perhaps the most studied form of cellular guidance and provides the foundation for connecting neurons throughout the nervous system[21]. Nascent myotubes, which are immature post-mitotic muscle precursors, also undergo cellular guidance and extend bipolar projections to connect with the tendon precursors that attach to the skeleton[22] (Supplementary Movie 1). The body wall muscles in *Drosophila* are easily visualized in live, unperturbed embryos[23,24], and have served as an essential model to understand the cellular and molecular mechanisms that direct muscle development[23–30]. The transmembrane receptors Heartless, Kon-tiki, and Robo regulate myotube guidance in *Drosophila* by transducing the navigational signals that direct myotube leading edges toward tendon precursors[23,28,30]. However, the intracellular pathways that link guidance receptors with the cytoskeletal changes underlying muscle morphogenesis are poorly understood.

Here we report that Polo regulates cellular morphogenesis in the post-mitotic mesoderm. The embryonic mesoderm undergoes multiple rounds of cell division after gastrulation, and mitosis in the mesoderm is largely complete by Stage 12. Nascent myotubes exit the cell cycle and begin myotube guidance during Stage 12, but surprisingly we found Polo is activated in the post-mitotic mesoderm during myotube guidance. Aurora kinases are the known activators of Polo, but the Aurora kinases were expressed in the embryonic mesoderm only until Stage 10. In contrast, the expression of the kinase Back seat driver (Bsd) was enriched in the Stage-12 mesoderm, where Bsd promoted Polo activation and directed microtubule reorganization necessary for myotube guidance. Thus the transition from mitosis to cellular morphogenesis is achieved through the spatially and temporally restricted expression of the Aurora kinases and Bsd. The Polo orthologue Plk1 was activated by the Bsd orthologue Vrk3 in mammalian cells, arguing Bsd regulates a conserved intracellular signaling pathway that directs muscle morphogenesis.

## Results

We carried out a forward genetic screen to identify regulators of myotube guidance, and uncovered a mutation in *CG8878* that disrupted muscle development. The body wall muscles in *CG8878* embryos showed pronounced navigational defects, so we named the gene *back seat driver* (*bsd*) (Fig. 1a–e and Supplementary Fig. 1B). The allele recovered from our screen (hereafter *bsd1*) is an embryonic lethal nonsense mutation (Q545*) that is predicted to produce a C-terminal truncation (Fig. 1c). *bsd* encodes a conserved serine/threonine kinase orthologous to the vaccinia-related kinases (VRKs), and proteins in the VRK family contain a single conserved kinase domain (KD) near the N-terminus, and a highly variable C-terminus (Fig. 1c and Supplementary Fig. 1C, D). The Bsd paralog Ballchen (Ball) regulates sarcomere assembly in adult flight muscles, but the null allele *ball2* is embryonic viable[31], suggesting Ball is not a major regulator of embryonic muscle development. Vertebrate VRK proteins have not been shown to regulate myogenesis, but pathogenic VRK1 variants have been identified in patients with motor neuropathies that may arise from defects in cellular morphogenesis[32].

**Bsd directs myotube guidance**. Thirty distinct muscles develop per embryonic hemisegment in *Drosophila*, and each muscle acquires a specific morphology (Fig. 1a, b). Myogenesis initiates with the specification of founder cells, which are muscle precursors with unique identities that form individual embryonic muscles. After specification, post-mitotic muscle founder cells break symmetry to begin elongation, and concurrently fuse with neighboring fusion competent myoblasts to form syncytial myotubes[25]. The nascent myotubes will elongate and identify muscle attachment sites on tendon cells in the ectoderm. The correct elongation and attachment of an individual muscle establishes a stereotypical morphology, and forms the largely invariant musculoskeletal pattern in abdominal segments A2-A8 (Fig. 1a, b). To broadly quantify morphology defects, we used an antibody against Tropomyosin to visualize muscle shape, and found a majority of the thirty muscles in each hemisegment showed significant morphology defects in *bsd1* embryos (Fig. 1a, b and Supplementary Fig. 1E).

To understand the myogenic role of Bsd in more detail, we used cell identity reporters to assay founder cell specification, myoblast fusion, and myotube guidance as previously described[23]. *5053.Gal4* is active in one founder cell that gives rise to the ventral lateral 1 (VL1) muscle, and *slou.gal4* is active in five founder cells that give rise to the dorsal transverse 1 (DT1), the longitudinal oblique 1 (LO1), the ventral acute 1 (VA1), the ventral acute 3 (VA3), and the ventral transverse 1 (VT1) muscles. *5053.Gal4* and *slou.gal4* expressing founder cells were correctly specified in *bsd* embryos (Supplementary Fig. 1F), but myotube elongation and muscle attachment site selection, which are the two hallmarks of myotube guidance, were significantly disrupted (Fig. 1d–g and Supplementary Movie 1). In contrast,

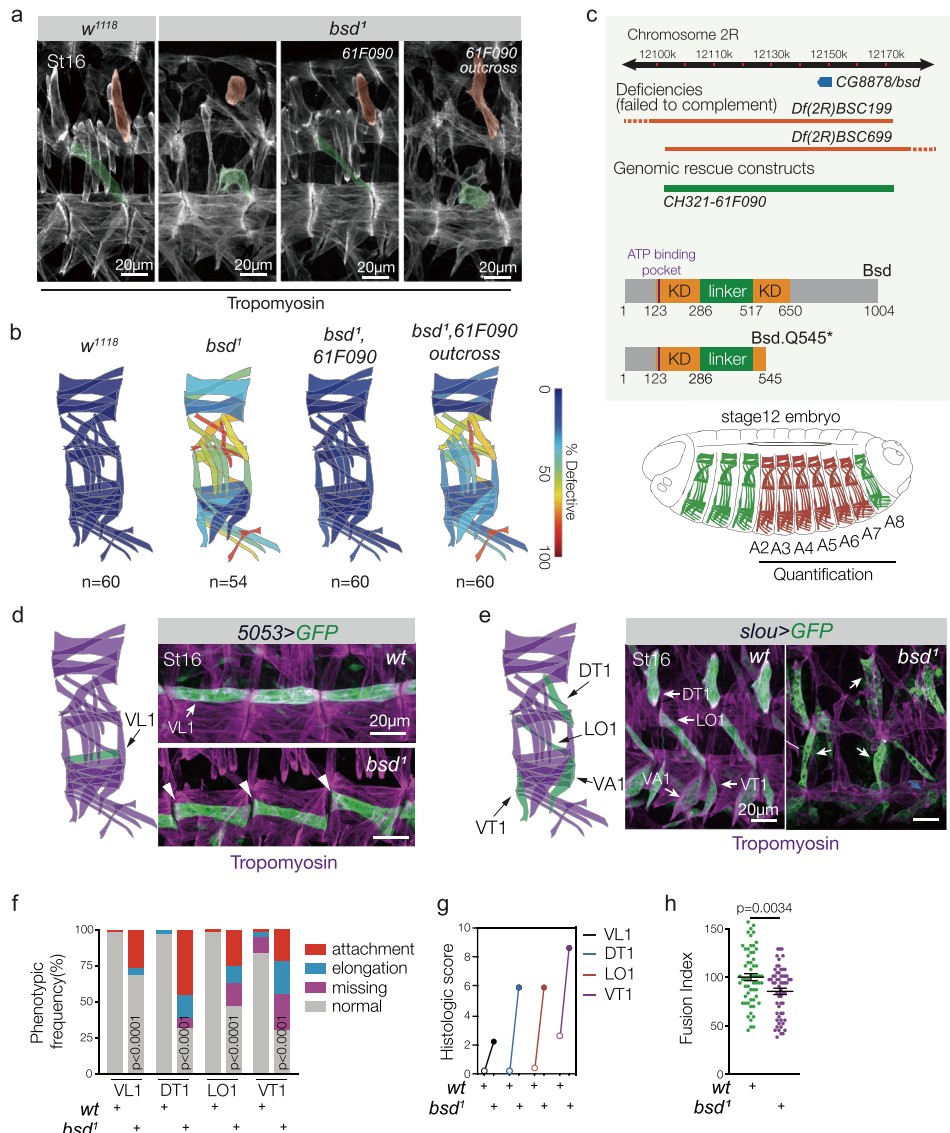

**Fig. 1 The ser/thr kinase Bsd regulates muscle morphogenesis. a** Myogenic phenotype in *bsd¹* mutants. Stage-16 embryos labeled with Tropomyosin. Doral transverse 1 (DT1) and longitudinal oblique (LO1) muscles are pseudocolored. Control (*w¹¹¹⁸*) embryos showed a stereotypic pattern of body wall muscles. *bsd¹* embryos showed severe defects in muscle morphology. *bsd¹* myogenic defects were completely rescued in embryos harboring the genomic construct CH321-61F090; outcrossing CH321-61F090 restored the *bsd¹* myogenic phenotype. **b** Quantification of muscle phenotypes. Muscles were scored in hemisegments A2–A7 of St16 embryos. Abnormal phenotypes (missing muscles, muscles with attachment site defects, and muscles that failed to elongate) were scored "defective". The frequencies of muscle defects are shown as a heatmap on the stereotypic muscle pattern in one embryonic hemisegment. (*n*) number of hemisegments scored. See Supplementary Fig. 1E for statistical analysis. **c** Mapping details and Bsd protein domains. Two overlapping deficiencies failed to complement *bsd¹* (dashed lines indicate breakpoints outside the genomic region shown; see Supplementary Fig. 1 for transheterozygous phenotypes). The Bsd protein has one kinase domain (KD; orange) that is conserved among the VRK protein family; the Bsd KD is divided by a unique linker (green). The position of a conserved ATP-binding pocket is also shown (purple). **d, e** Stage-16 embryos labeled for *5053 > GFP* (**d**, green) or *slou > GFP* (**e**), and Tropomyosin (violet). Ventral lateral 1 (VL1) muscles made incorrect or incomplete tendon attachments in *bsd¹* embryos (**d**, white arrowheads) and were often rounded (not shown). DT1, LO1, and ventral transverse 1 (VT1) muscles also made incorrect tendon attachments in *bsd¹* embryos (**e**, white arrowheads; see Supplementary Movie 1) or were rounded (blue arrowhead). **f** Histogram of muscle phenotypes in *5053 > GFP* and *slou > GFP* embryos (*n* = 60 hemisegments per muscle per genotype). **g** Histologic scores for muscles analyzed in (**f**). Open circles (wild-type), closed circles (*bsd¹*). **h** Fusion index in St12 embryos. Myoblast fusion was modestly reduced in *bsd¹* embryos. Each data point represents one hemisegment (*n* = 60 per genotype). See "Methods" for statistical parameters. Significance determined by two-sided Fisher's exact test (**b**, **f**), and two-sided unpaired Student's *t* test (**h**). Error bars represent SEM.

tendon cells developed normally in *bsd* embryos (Supplementary Movie 2), suggesting the myotube guidance defects we observed are not secondary to tendon cell fate specification or tendon cell positioning. We also found that the initial round of myoblast fusion was modestly reduced in *bsd* embryos (*bsd* fusion index = 85.7%; Fig. 1h), although not to the degree reported for other well-characterized fusion mutants such as *loner*[29]. Bsd is

thus an essential regulator of muscle morphogenesis that directs myotube guidance and, to a lesser extent, myoblast fusion.

**Bsd acts cell-autonomously in the mesoderm**. We generated and validated an antibody against Bsd (Supplementary Fig. 2A–C and Supplementary Data 3), and found Bsd was ubiquitously expressed in

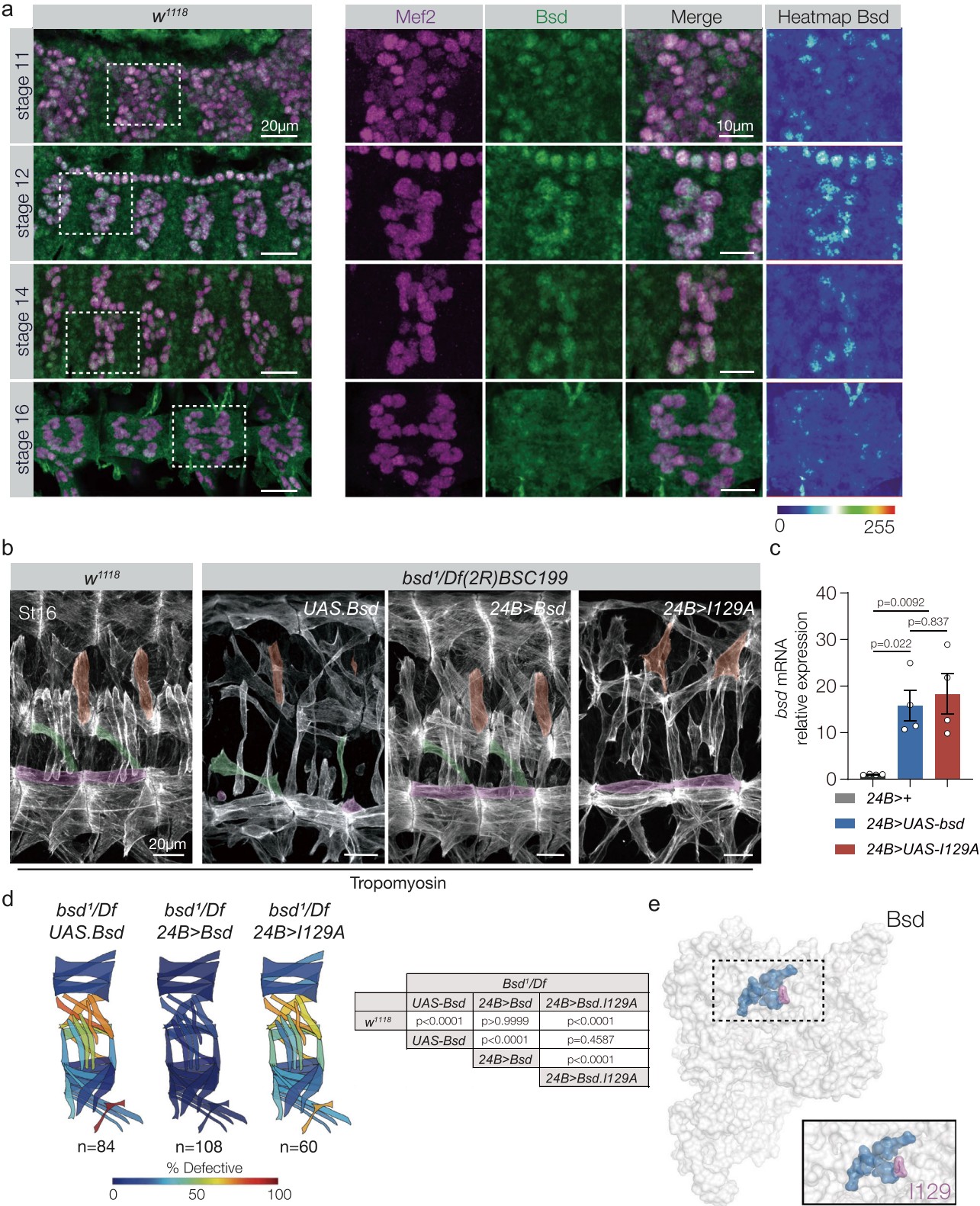

blastoderm embryos (Supplementary Fig. 2D). However, after gastrulation, Bsd expression was more dynamic. Founder cells are specified during Stages 10–11, and Bsd expression in the mesoderm during Stage 11 was reduced compared to subsequent stages of myogenesis (Fig. 2a and Supplementary Fig. 2D–F). Myotube guidance begins at Stage 12, and Bsd expression levels in the mesoderm

peaked during Stages 12–14 (Fig. 2a and Supplementary Fig. 2D–F). Myotube guidance is complete by Stage 16, and the relative levels of Bsd expression in the mesoderm were reduced in Stage-16 embryos (Fig. 2a and Supplementary Fig. 2D–F). The Bsd expression pattern is consistent with our hypothesis that kinase expression is spatially and temporally regulated during embryogenesis.

**Fig. 2 Bsd expression in the mesoderm is progressively enriched during muscle development. a** $w^{1118}$ embryos immunolabeled for Bsd (green) and Mef2 (violet). Bsd was detected in Mef2+ muscle cells throughout myogenesis. Bsd protein levels in the mesoderm peaked during Stage 12. Bsd also appeared to be enriched in myonuclei (see Supplementary Fig. 2E, F). **b** Bsd kinase activity in the mesoderm directs muscle morphogenesis. Control ($w^{1118}$), $bsd^1/Df(2 R)$ BSC199 UAS.Bsd, $bsd^1/Df(2 R)BSC199$ 24B.Gal4 > Bsd, and $bsd^1/Df(2 R)BSC^199$ 24B.Gal4 > Bsd.I129A embryos labeled for Tropomyosin. DT1, LO1, and VL1 muscles are pseudocolored orange, green, and violet. $bsd^1$ embryos that expressed wild-type Bsd in the mesoderm showed improved muscle morphology; $bsd^1$ embryos that expressed catalytically inactive Bsd.I129A showed extensive muscle defects. **c** Quantitative real-time PCR analysis of bsd mRNA showed equivalent expression between transgenic lines (for each group, n = 4). **d** Quantification of muscle phenotypes as described in Fig. 1b. **e** Space-filling model of Bsd. The predicted ATP-binding pocket is colored with Iso129 shaded violet. Significance was determined by one-way ANOVA (**c**) and two-sided Fisher's exact test (**d**). Error bars represent SEM.

The temporally dynamic expression of Bsd in the mesoderm suggested Bsd functions cell-autonomously to regulate muscle morphogenesis. $bsd^1$ embryos that expressed wild-type Bsd in the mesoderm under the control of 24B.Gal4 showed a largely normal muscle pattern (Fig. 2b–d). Proteins in the VRK family contain a highly conserved ATP-binding pocket that is essential for catalytic activity ((Supplementary Fig. 1D) and ref. [33]), and our computational structural models predicted that Bsd isoleucine 129 binds ATP (Fig. 2e). $bsd^1$ embryos that expressed Bsd.I129A in the mesoderm showed a myogenic phenotype that was indistinguishable from $bsd^1$ embryos (Fig. 2b–d), arguing Bsd kinase activity is required for proper muscle morphogenesis.

**Bsd activates Polo to direct muscle morphogenesis.** To uncover Bsd effectors during myotube guidance, we used recombinant Bsd to precipitate Bsd-interacting proteins from whole embryo lysates, which were then sequenced by Mass Spectrometry (MS; Fig. 3a and Supplementary Fig. 3A). The MS results identified over 150 candidates that interacted with Bsd, none of which were known to regulate muscle morphogenesis (Supplementary Data 2). Since the absolute quantity of each candidate in the input embryo lysate was not known, the relative abundance of a candidate in the pool of precipitated proteins could not be used to prioritize the potential Bsd-interacting proteins. To rank the candidates for further analysis, we first validated protein–protein interactions in S2 cells, and then characterized muscle morphogenesis in embryos with reduced candidate gene function. The mitotic kinase Polo showed the strongest interaction with Bsd in S2 cells (Supplementary Fig. 3B), and we confirmed the interaction with reciprocal immunoprecipitation experiments (Fig. 3b, c). Strikingly, the Bsd.Q545* protein encoded by our EMS-induced $bsd^1$ allele did not interact with Bsd (Fig. 3b, c). Embryos homozygous for the hypomorphic alleles $polo^1$ and $polo^{KG03033}$ showed myogenic phenotypes similar to $bsd$ embryos (Supplementary Fig. 3C, D). We tested four candidates in addition to Polo, and although these proteins showed a weak interaction with Bsd in S2 cells (Supplementary Fig. 3B), we could not confirm muscle phenotypes with alleles that affected the remaining candidates. Polo was thus the highest-priority candidate for further analysis.

Germline clones of the hypomorphic allele $polo^1$ reduce the maternal contribution of Polo, and $polo^1$ maternal mutant embryos showed defects in mitotic chromosome alignment[34]. However, $polo^1$ zygotic mutants with a normal maternal contribution of Polo were largely viable, and did not show embryonic mitotic defects[35]. After gastrulation, the mesoderm undergoes four rounds of cell division that conclude by Stage 11[36]. During Stages 10–11, each founder cell is specified from a pool of cells known as an equivalence group. The remaining cells in the equivalence group then differentiate into fusion competent myoblasts[25]. To understand if $polo^1$ zygotic mutant embryos undergo normal cell divisions prior to founder cell specification, we quantified the number of Mef2-positive somatic mesoderm cells in Stage 10 embryos. Although $polo^1$ embryos showed a 14.2% decrease in the number of Mef2-positive cells compared to wild-type embryos (Supplementary Fig. 3E, F), the number of 5053.Gal4 and slou.gal4 expressing founder cells were not significantly different between wild-type and $polo^1$ Stage-12 embryos (Supplementary Fig. 3G, H). These observations suggest that the mitotic defects in $polo^1$ zygotic mutant embryos do not appreciably affect equivalence group size or founder cell specification. Using 5053.Gal4 and slou.gal4 to characterize muscle morphogenesis, we identified myotube elongation and muscle attachment site selection defects in the VL1, DT1, LO1, and VT1 muscles of $polo^1$ embryos (Fig. 3d, e), suggesting Polo directs myotube guidance after founder cell specification. Importantly, the severity of muscle morphogenetic defects was equivalent among $bsd^1$, $polo^1$, and $bsd^1$ $polo^1$ embryos, arguing Bsd and Polo act in a common pathway to direct myotube guidance (Fig. 3f–h and Supplementary Fig. 3I).

Since Bsd and Polo are both protein kinases, we used phosphorylation assays to understand if one protein might act upstream of the other. A common strategy for measuring protein phosphorylation is to immunoprecipitate target proteins, and assay total phosphorylation by western blot. Bsd is a predicted serine/threonine kinase, and Polo isolated from S2 cells transfected with wild-type Bsd had significantly more phosphorylated threonine than controls (Fig. 4a and Supplementary Fig. 4A). However, Polo isolated from cells transfected with kinase-dead Bsd.I129A did not show a significant change in the threonine phosphorylation (Supplementary Fig. 4B). Polo is activated by phosphorylation of Thr182[17], so we repeated our phosphorylation assay with an antibody that specifically recognizes Polo$^{pThr182}$ and found Bsd significantly increased Polo$^{pThr182}$ levels (Supplementary Fig. 4C, D). Activated Polo$^{pThr182}$ translocates to the nucleus[20], and Bsd promoted the nuclear translocation of wild-type Polo but not phospho-dead Polo.T182A in S2 cells (Fig. 4b, c). These S2 cell studies suggested Bsd phosphorylates Polo, so we performed cell-free kinase assays and found Bsd phosphorylated Polo on Thr182 (Supplementary Fig. 4E). Together our data argue that Polo is a direct substrate of Bsd kinase activity.

To understand how Polo activity is regulated in vivo, we assayed Polo phosphorylation in whole embryo lysates and found Bsd broadly promoted Polo phosphorylation on threonine residues (Fig. 4d). Similar to Bsd expression (Fig. 2a), activated Polo$^{pThr182}$ levels were dynamic in the mesoderm, and significantly increased during Stage 12 (Fig. 4e and Supplementary Fig. S4F). However, in $bsd^1$ embryos, Polo$^{pThr182}$ levels did not change during Stage 12 (Fig. 4e and Supplementary Fig. 4F), despite the fact that Polo protein levels were comparable between wild-type and $bsd^1$ embryos (Supplementary Fig. 4F). Functionally, $bsd^1$ embryos that expressed activated Polo.T182D in the mesoderm under the control Mef2.Gal4 showed improved muscle morphogenesis compared to $bsd^1$ embryos (Fig. 4g). Bsd therefore directs Polo activation in the mesoderm of Stage-12 embryos, and Bsd-mediated Polo activation is required for myotube guidance.

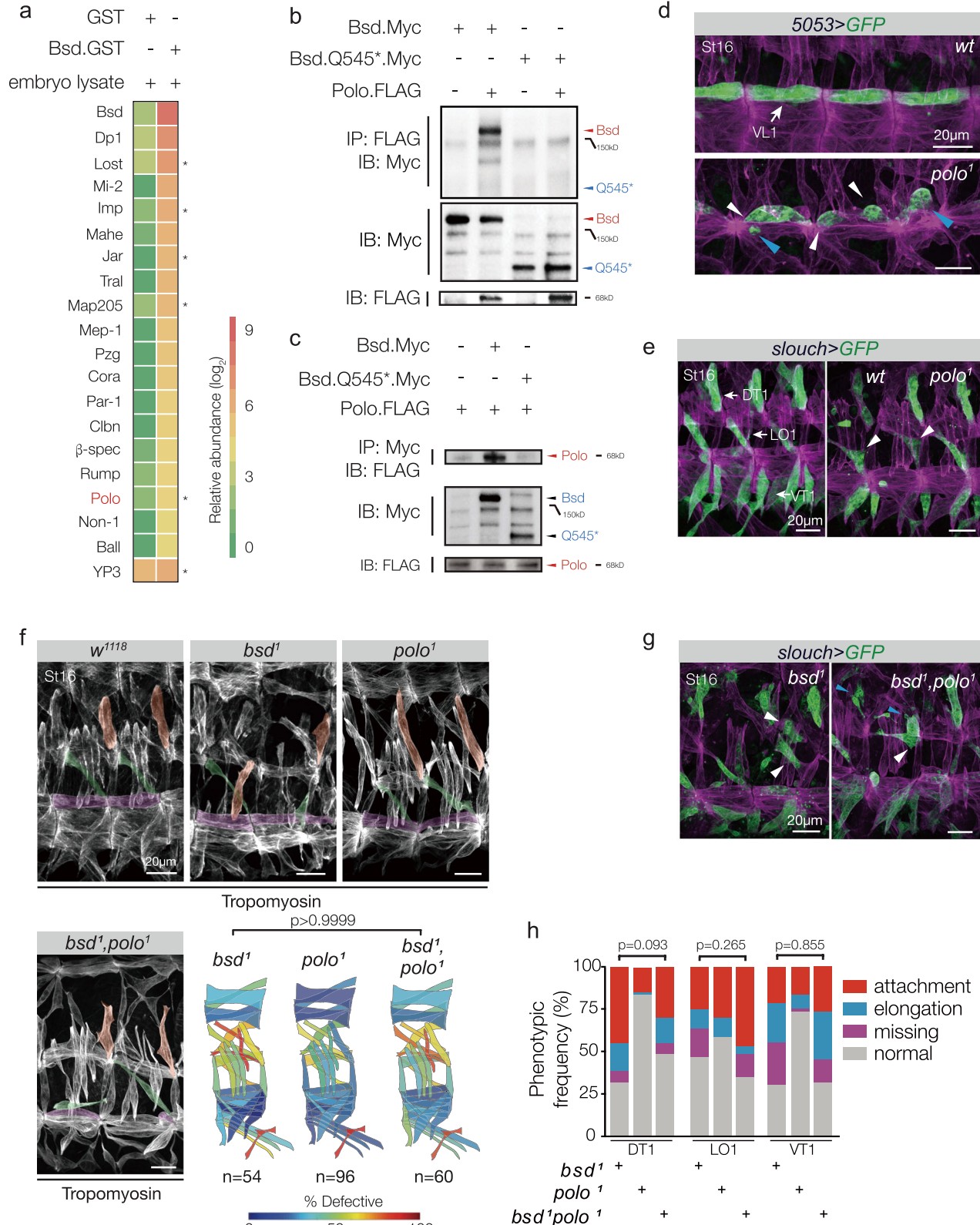

**Aurora kinases are not active in the post-mitotic mesoderm.** Nascent myotubes exit the cell cycle and begin myotube guidance during Stage 12 (Supplementary Fig. 5A), so it was surprising that a mitotic kinase-like Polo was activated in the post-mitotic mesoderm (Supplementary Fig. 4F). The pro-mitotic Aurora kinases are the only known activators of Polo or its vertebrate orthologue Plk1[20,37]. *aurora A* (*aurA*) and *aurora B* (*aurB*) were broadly expressed in the mitotic mesoderm until Stage 10, but by Stage 12 *aurA* and *aurB* expression was undetectable in the somatic mesoderm (Fig. 5a–c and Supplementary Fig. 5A, B). Alternatively, Bsd expression was enriched in the mesoderm during Stage 12 (Fig. 2a), suggesting the control of Polo activation transitions from the Aurora kinases to Bsd during myotube guidance. To test this model, we assayed Polo[pThr182] levels in the

**Fig. 3 Bsd and Polo kinase act in a common myogenic pathway. a** Relative abundance of Bsd-interacting proteins recovered from 12 to 24 h embryo lysates, determined by MS. See Supplementary Table S2 for detailed quantification. **b, c** Immunoprecipitation of S2 cell lysates transfected with Bsd, Bsd.Q545*, and Polo. Full-length Bsd interacted with Polo in reciprocal experiments, but Bsd.Q545* and Polo did not interact. **d** polo[1] VL1 muscle phenotype. Stage-16 embryos labeled for 5053 > GFP (green) and Tropomyosin (violet). polo[1] VL1 muscles were rounded (blue arrowhead) and made incorrect or incomplete tendon attachments (white arrowheads). **e** polo[1] DT1, LO1, and VT1 muscle phenotypes. Stage-16 embryos labeled for slou > GFP (green) and Tropomyosin (violet). polo[1] LO1 muscles had attachment site defects (white arrowheads). **f** Muscle phenotypes in bsd[1] polo[1] double mutants. Stage-16 embryos labeled with Tropomyosin. DT1, LO1, and VL1 muscles are pseudocolored orange, green, and violet. The frequency and severity of muscle morphology defects was comparable between bsd[1] and bsd[1] polo[1] embryos. Quantification of muscle phenotypes is as described in Fig. 1b (see Supplementary Fig. 3G for statistical analysis). **g** DT1, LO1, and VT1 phenotypes in bsd[1] polo[1] double-mutant embryos (labeled as in **e**). The frequency and severity of DT1, LO1, and VT1 phenotypes were comparable between bsd[1] and bsd[1] polo[1] embryos. **h** Histogram of VL1, DT1, LO1, and VT1 phenotypes from 5053 > GFP and slou > GFP embryos (bsd[1], n = 60; polo[1], n = 60; bsd[1];polo[1], n = 60). Significance was determined by two-sided unpaired Student's t test (**a, d**), one-way ANOVA (**c, f**), and two-sided Fisher's exact test (**g**).

mesoderm of Stage-12 aurA and aurB embryos, and found Polo[pThr182] levels were not significantly different between mutant and control embryos (Fig. 5d). In addition, aurA and aurB embryos showed normal muscle morphology at the end of embryogenesis (Supplementary Fig. 5C, D). These data further suggest the control of Polo activation transitions from the Aurora kinases to Bsd during myotube guidance.

We next asked why it might be necessary for Bsd to regulate Polo activity in place of AurA and AurB. The Aurora kinases regulate multiple mitotic proteins in addition to Polo/PLK1[38], and we hypothesized that Aurora kinase expression is reduced after Stage 10 to promote cell cycle exit in the mesoderm. We used Mef2.Gal4 to temporally misexpress aurB in the mesoderm, and found Mef2 > aurB embryos had significantly more mitotic cells than controls (Supplementary Fig. 5E–G). AurB specifically promotes the transition from S phase to the G2/M phase of the cell cycle[39], and Mef2 > aurB embryos had significantly more mesoderm cells in the G2/M phase than control embryos (Fig. 5e, f). Functionally, Mef2 > aurB muscles had more myonuclei at the end of myogenesis than controls (Supplementary Fig. 5H, I), arguing extended mitosis in the mesoderm produced more myogenic precursors. Thus the control of Polo activity during myotube guidance transitions away from the Aurora Kinases to promote cell cycle exit.

**Polo effectors Tum and Pav direct myotube guidance.** How then does Bsd-activated Polo regulate myogenesis? Two Drosophila Polo effector proteins, the GTPase activating protein Tumbleweed (Tum) and the kinesin microtubule motor protein Pavarotti (Pav), coordinate cytoskeletal dynamics to position furrow formation at the onset of cytokinesis[18,40,41]. Polo directly interacts with Tum, Tum directly interacts with Pav, and Polo is required for the correct localization of Tum and Pav during cytokinesis[18,42]. In addition, a role for Tum in myotube guidance was suggested in studies showing Tum regulates the microtubule cytoskeleton to direct myotube elongation[27]. The Polo/Tum/Pav cytoskeletal regulatory module that directs cytokinesis may therefore be reactivated in the post-mitotic mesoderm to direct microtubule reorganization during myotube guidance. To extend previous work, we reanalyzed muscle morphogenesis in tum embryos using the cell identity reporters 5053.Gal4 and slou.gal4, and found Tum directs both myotube elongation and muscle attachment site selection during myotube guidance (Fig. 6a, b and Supplementary Movie 3). In addition, the severity of muscle morphogenetic defects was equivalent among polo[1], tum[DH15], and polo[1] tum[DH15] embryos (Supplementary Fig. 6A, B). pav[B200] embryos also showed muscle phenotypes, and the severity of muscle morphogenetic defects was equivalent among tum[DH15], pav[B200], and tum[DH15] pav[B200] embryos (Supplementary Fig. 6C).

These genetic studies suggest Polo, Tum, and Pav act in a common myogenic pathway, and argue Bsd reactivates the Polo/Tum/Pav cytoskeletal regulatory module to direct myotube guidance.

**Bsd regulates the microtubule cytoskeleton.** The microtubule cytoskeleton transitions from a cortical organization in founder cells to a linear organization in nascent myotubes that parallels the axis of elongation (Supplementary Movie 4)[27]. Live imaging of microtubule reorganization revealed that the microtubule transition was delayed by over 60 min in bsd myotubes, and that bsd myotubes failed to maintain linear microtubule arrays (Fig. 6c, d and Supplementary Movie 4). However, the actin cytoskeleton was largely unaffected in bsd myotubes (Supplementary Fig. 6D). The microtubule minus-end nucleator γ-tubulin initiates the assembly of new microtubules[43], and at the onset of myotube elongation γ-tubulin foci are predominantly localized to the myotube cortex[27]. As the microtubule cytoskeleton transitions to linear arrays, γ-tubulin foci appear in the internal myotube cytoplasm through a Tum-dependent mechanism (Fig. 6e, f)[27]. In bsd[1] embryos, γ-tubulin foci failed to accumulate in the myotube cytoplasm, suggesting Tum-dependent γ-tubulin mediated microtubule nucleation is defective in bsd myotubes (Fig. 6e, f and Supplementary Fig. 6E, F). Bsd is thus an essential regulator of the microtubule cytoskeleton.

**Bsd and Polo orthologues regulate muscle morphogenesis.** To understand if the regulatory functions of Bsd are conserved, we used small interfering RNAs (siRNAs) to knock down Vrk1, Vrk2, and Vrk3 during mammalian muscle morphogenesis. Under culture conditions that promote differentiation, C2C12 cells (immortalized mouse myoblasts) will form nascent myotubes that extensively elongate[44]. C2C12 cells treated with Vrk1 and Vrk2 siRNAs were morphologically similar to control-treated cells after 7 days of differentiation (Supplementary Fig. 7A, B), but C2C12 cells treated with Vrk3 siRNAs showed significantly reduced elongation and a reduced fusion index (Fig. 7a–d and Supplementary Fig. 7A–D). These myogenic assays functionally confirmed our phylogenetic analysis, arguing Bsd is most similar to Vrk3 (Supplementary Fig. 1C). Post-mitotic C2C12 cells treated with the Plk1 inhibitor Volasterib phenocopied C2C12 cells treated with Vrk3 siRNAs (Fig. 7a–d). In addition, Vrk3 physically interacted with Plk1 (Fig. 7e) and promoted activating phosphorylation of Plk1 in HEK293 cells (Fig. 7f). Thus the Bsd orthologue Vrk3 activates the Polo orthologue Plk1 in mammalian cells. In addition, bsd[1] embryos that expressed Vrk3 in the mesoderm showed largely normal muscle morphology (Fig. 7g). Our results argue that the functions of Bsd and Vrk3 are highly conserved.

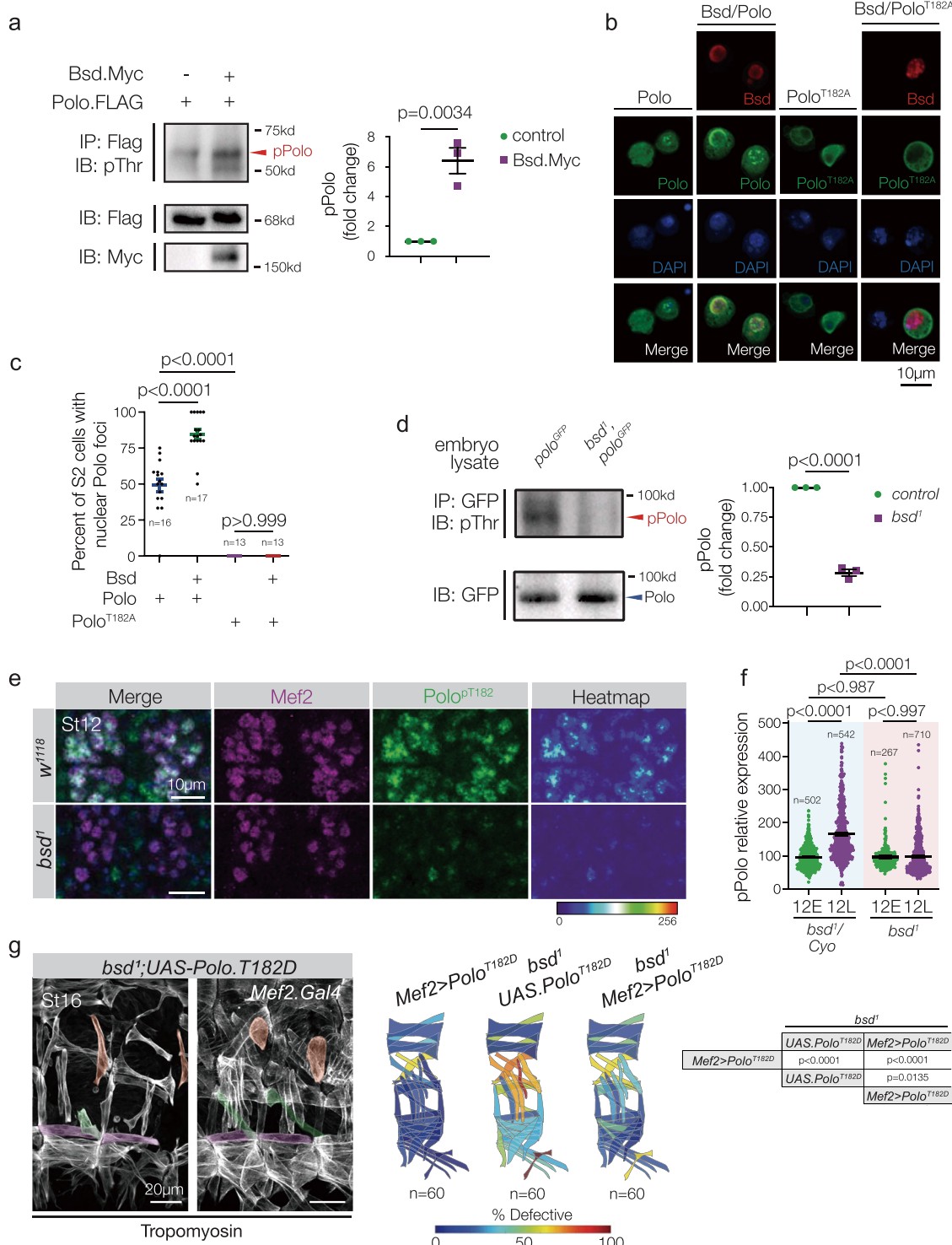

## Discussion

This study identified Bsd as a conserved regulator of Polo activation. Bsd promoted Polo phosphorylation in cultured cells and in cell-free kinase assays (Fig. 4a), and Bsd was required to activate Polo in the post-mitotic mesoderm (Fig. 4e and Supplementary Fig. 4E). In addition, activated Polo (Polo.T182D) rescued the *bsd* myogenic phenotype (Fig. 3f), which argues an essential function of Bsd during myotube guidance is to activate Polo. The microtubule cytoskeleton reorganizes to drive

morphological changes in nascent myotubes, and Bsd directed microtubule reorganization during myotube guidance (Fig. 6c–f and Supplementary Movie 4). These observations are consistent with a model in which Bsd activates a Polo/Tum/Pav cytoskeletal regulatory module to direct cellular morphogenesis (Fig. 7h and Supplementary Fig. 7E).

We have also identified an example in which the dynamic expression of regulatory kinases directs the transition from mitosis to cellular morphogenesis. The Aurora kinases are

**Fig. 4 Bsd activates Polo to direct myotube guidance. a** Polo immunoprecipitated from S2 cell lysates and blotted with anti-phosphothreonine (pThr). Cells transfected with Bsd showed significantly more phosphorylated Polo than controls. (for each group, $n = 3$). **b** Bsd.Myc, Polo.Flag, and Polo.T182A.Flag were transfected into S2 cells; transgenic proteins were detected with anti-Myc (red, Bsd) and anti-Flag (green, Polo). Polo localized to the nucleus in a subset of control cells (left column). The frequency of cells showing nuclear Polo localization was increased in cells co-transfected with Bsd (second column from left). Inactivatable Polo (T182A) did not localize to the nucleus in control cells (third column from left) or in cells co-transfected with Bsd (right column). **c** Dot plot showing the percent of cells with nuclear Polo per microscope field from (**b**). **d** Endogenous GFP-tagged Polo was immunoprecipitated from embryo lysates and blotted with pThr. $bsd^1$ embryo lysates showed significantly less phosphorylated Polo than controls (for each group, $n = 4$). **e** Stage-12 embryos immunolabeled for Polo phosphorylated at Thr182 (Polo$^{pT182}$, green) and Mef2 (violet). Polo$^{pT182}$ levels were reduced in the myonuclei of $bsd^1$ embryos compared to controls. **f** Polo$^{pT182}$ fluorescent intensity from embryos shown in (**d**). Polo$^{pT182}$ fluorescence intensity is dynamic during Stage 12, and significantly increased in control ($bsd^1$/Cyo) embryos. Polo$^{pT182}$ fluorescence intensity did not increase in Stage 12 $bsd^1$ embryos. Control and experimental embryos were derived from the same preparation. **g** Activated Polo rescues the $bsd$ phenotype. $bsd^1$ UAS.Polo$^{T182D}$ and $bsd^1$ Mef2.Gal4 > Polo$^{T182D}$ embryos labeled for Tropomyosin. DT1, LO1, and VL1 muscles are pseudocolored orange, green, and violet. $bsd^1$ embryos that expressed active (phosphomimetic) Polo.T182D in the mesoderm showed improved muscle morphology compared to $bsd^1$ embryos. Note that expressing Polo.T182D in otherwise wild-type embryos caused a modest muscle phenotype, which may explain the incomplete rescue of the $bsd^1$ phenotype. Quantification of myogenic phenotypes is as described in Fig. 1b. 12E Stage 12 early, 12L Stage 12 late. Significance was determined by two-sided unpaired Student's $t$ test (**a**, **d**), one-way AONVA (**c**, **f**), and two-sided Fisher's Exact test (**g**). Error bars represent SEM.

broadly expressed in the mitotic mesoderm, where they presumably activate Polo and other target proteins to initiate mitotic entry and complete the critical steps of mitosis (Fig. 5a). The Aurora kinases are not expressed in the post-mitotic mesoderm, and the temporal misexpression of AurB prolonged cell division in the mesoderm (Fig. 5e, f). Bsd expression was complementary to the Aurora kinases (Figs. 2a and 5a–c), and Bsd-activated Polo in the post-mitotic mesoderm to direct muscle morphogenesis (Fig. 4e–g). Over 50% of the *Drosophila* kinases with known embryonic expression patterns transition from ubiquitous expression before gastrulation to tissue-specific expression after gastrulation, and an additional 20% of kinases show spatially restricted expression throughout development (Supplementary Data 1 and Supplementary Fig. 1A). Zebrafish kinases show similar embryonic expression patterns (Supplementary Data 1 and Supplementary Fig. 1A). The dynamic expression of protein kinases in both invertebrates and vertebrates suggest that expression-based regulation of kinase signaling pathways could be broadly employed to direct the key events of embryogenesis.

Polo was reactivated in the post-mitotic mesoderm (Supplementary Fig. 4F), which suggests Polo activity alone is not sufficient for mesodermal cells to enter mitosis. On the other hand, ectopic AurB expression in the mesoderm increased the proportion of cells in the G2/M phase of the cell cycle (Fig. 5e, f). Since AurB and Bsd activate Polo by phosphorylating Thr182 (Fig. 4e)[19,20], AurB likely induces cell cycle progression through a Polo Thr182-independent mechanism. Cyclins are key regulators of the cell cycle, and the regulation of cyclin activity is best understood at the level of protein expression. However, the G2/M checkpoint regulator Cyclin B (CycB) can be detected throughout the cell cycle[45]. At the G2/M checkpoint CycB activity is phosphorylation-dependent, where active CycB is phosphorylated on residues near the nuclear export sequence and also dephosphorylated by Cdc25 on residues near the N-terminus[46]. In the *Drosophila* germline AurB promotes the phosphorylation of CycB[45], and in human lymphoma cells AurB increases the proportion of cells in G2/M[39]. AurA also activates CDC25[38], which further implicates Aurora kinases as Polo/PLK1-independent regulators of cell cycle progression. Our studies argue that the control of the Polo transitions from the Aurora kinases to Bsd so that the Polo/Tum/Pav cytoskeletal regulatory module can be reactivated in differentiating, post-mitotic myotubes without promoting mitotic entry.

Bsd regulates myotube elongation and muscle attachment site selection (Supplementary Movie 1), which are the hallmarks of myotube guidance. Our myotube guidance hypothesis argues that extracellular guidance cues, such as FGFs, corroborate with contact-dependent interactions between myotubes and tendons to pattern the musculoskeletal system[23,47]. The Bsd paralog Ball is phosphorylated during female meiosis where it functions to maintain spindle integrity[48]. Similar to Ball, Bsd might be phosphorylated and activated by an upstream kinase. Our MS experiment identified Par-1, Target of Rapamycin, and SR Protein Kinase (SRPK) as potential upstream kinases that could activate Bsd (Supplementary Data 2). Mammalian SRPK1 was identified in a screen for tumor cell migration[49], and has been implicated in breast, lung, and renal cancer metastasis suggesting SRPKs promote the cytoskeletal changes necessary for cellular migration and cellular guidance[49–51]. Although *Drosophila* SRPK has not been linked to the extracellular receptors known to regulate myotube guidance, murine *Srpk3* is a Mef2 target gene in striated muscle[52], and SRPK proteins regulate developmental signaling pathways[53]. We favor a model in which an intracellular effector, such as SRPK, transduces guidance cues from the cell surface to activate Bsd and direct myotube elongation toward muscle attachment sites.

We found that activated Polo partially rescued the $bsd^1$ embryonic phenotype (Fig. 4g). While the levels of activated Polo in the rescue experiment may not have recapitulated endogenous levels of activated Polo, it is also possible that Bsd regulates Polo-independent pathways. For example, we validated a physical interaction between Bsd and the Myosin VI protein Jaguar (Jar; Supplementary Fig. 3B). Jar stabilizes interactions between the actin and microtubule cytoskeletons[54], and is required for cell migration[55] and cellular guidance[56]. Although one essential role for Bsd is to activate the Polo/Tum/Pav cytoskeletal regulatory module, it seems likely that Bsd regulates additional cytoskeletal proteins, such as Jar, to direct myotube guidance.

Polo directs spindle formation, cytokinesis, and myotube guidance by regulating the microtubule cytoskeleton. The *polo^1* allele was originally identified as a maternal effect mutation that disrupted mitotic chromosome alignment[34], and Polo was later found to recruit γ-tubulin and Abnormal spindle protein (Asp) to the centrosome for microtubule nucleation[57]. The role of Polo during spindle assembly appears to be Tum and Pav independent, but during cytokinesis the Polo/Tum/Pav regulatory module uses the central spindle microtubules to position the contractile ring and initiate furrow formation[18,58,59]. We found that Bsd directs microtubule reorganization during myotube guidance, which may involve the recruitment of γ-tubulin to sites of microtubule

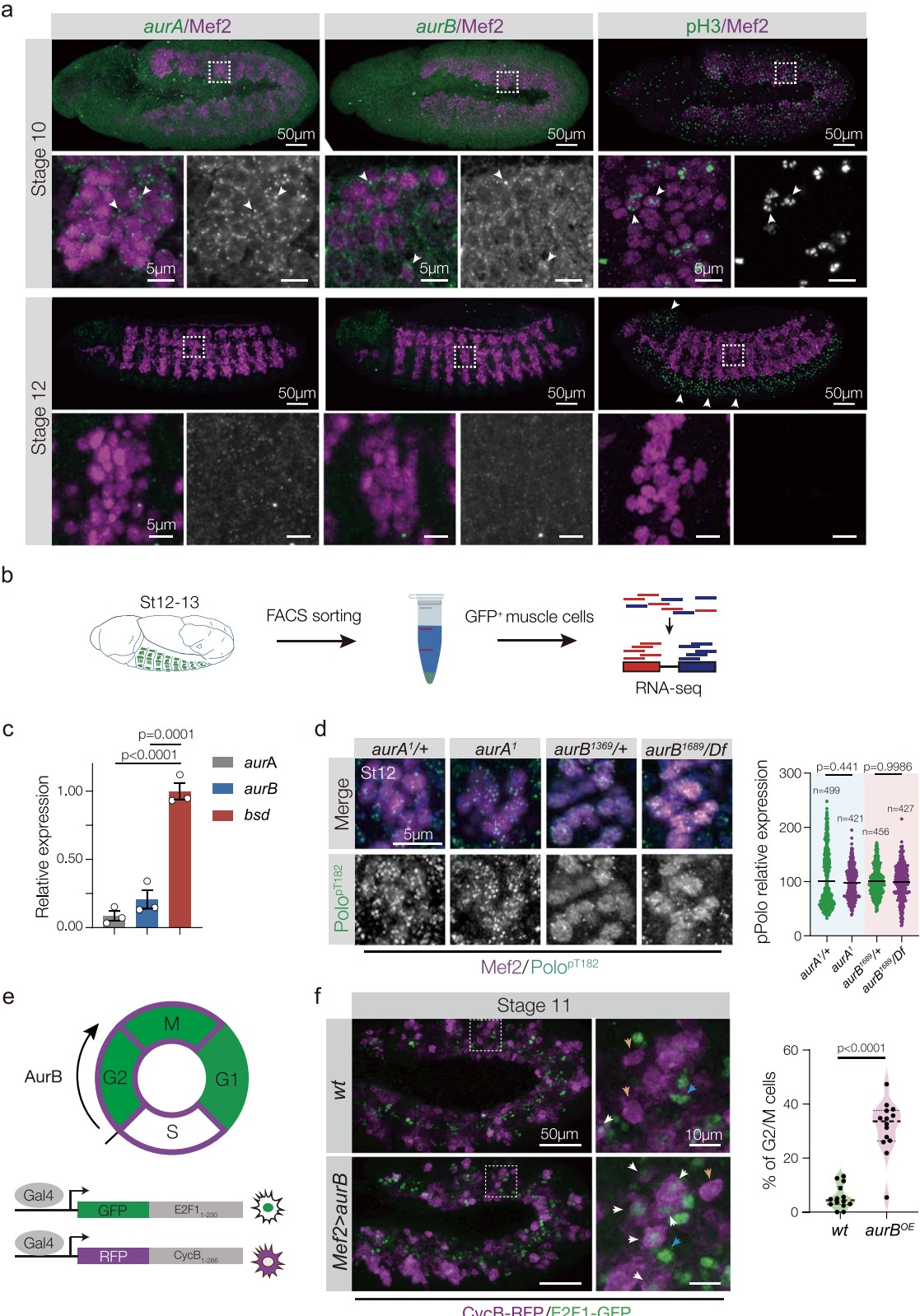

nucleation (Fig. 6c–f). Tum also regulates myotube guidance (Fig. 6a, b), and is required for γ-tubulin localization in post-mitotic myotubes[27]. The precise mechanism by which Bsd and the Polo/Tum/Pav cytoskeletal regulatory module interact to direct myotube guidance remains unclear, but Tum and Pav appear to function outside of the nucleus to direct changes to the microtubule cytoskeleton in post-mitotic cells[27]. One possibility is that Bsd-activated Polo translocates to the myonucleus to direct

**Fig. 5 Aurora kinases are inactive in the post-mitotic mesoderm. a** Stage 10 and Stage-12 embryos labeled for *aurA* (left, green), *aurB* (middle, green) or pH3 (right, green) and Mef2 (violet). *aurA*, *aurB* and pH3 are expressed in the mesoderm of Stage 10 embryos, but are excluded from the somatic mesoderm in Stage-12 embryos. **b** Nascent myotubes that expressed *rp298 > eGFP* were sorted to enrich for myogenic RNA, and RNA-seq results were reported in ref. [23]. **c** Analysis of *aurA*, *aurB* and *bsd* mRNA expression in nascent myotubes from the experiment shown in (**b**) (for each group, $n = 4$). **d** Stage-12 embryos immunolabeled for activated Polo (Polo$^{PT182}$, green) and Mef2 (violet). Polo$^{PT182}$ levels were comparable between *aurA*, *aurB*, and control embryos. Control and experimental embryos were derived from the same preparations. **e** The Fly-FUCCI system. A tissue-specific Gal4 activates fluorescent fusion proteins that are post-transcriptionally controlled to identify cell cycle phases in vivo. **f** Stage 11 embryos that expressed Fly-FUCCI transgenes under the control of *Mef2.Gal4*, immunolabeled for GFP-E2F1$_{1-230}$ (green) and RFP-CycB$_{1-266}$ (violet). Embryos that expressed *aurB* in the mesoderm showed more cells in G2/M (white arrowheads) than G1 (blue arrowheads) or S (orange arrowheads). Percent of dividing cells in G2/M phase (GFP/RFP double-positive cells) are plotted, the median and interquartile ranges are shown (dotted horizontal lines); each data point represents a single hemisegment.(*wt*, $n = 14$; *aurB*$^{OE}$, $n = 14$). Significance was determined by one-way ANOVA (C) and two-sided unpaired Student's *t* test (**d**, **f**). Error bars represent SEM.

the nuclear export of Tum and Pav after founder cells and fusion competent myoblasts have exited the cell cycle. Cytoplasmic Tum/Pav complexes would in turn reorganize the microtubule cytoskeleton to direct myotube elongation. Tum and Pav also regulate axon guidance in post-mitotic neurons[60], so understanding if Polo directly phosphorylates Tum and Pav to regulate the subcellular localization of the Tum-Pav complex may inform our overall understanding of cellular guidance. We predict that the Polo/Tum/Pav cytoskeletal regulatory module functions in many cell types to regulate tissue morphogenesis.

Mammalian Vrk3 regulated myotube elongation, physically associated with Plk1, and promoted activating phosphorylation of Plk1 (Fig. 7a–f). The active site in human VRK3 is divergent at three residues, which led to the hypothesis that VRK3 is a pseudokinase[61]. However, subsequent studies show that VRK3 has kinase activity under certain contexts[62], and our studies support an active role for Vrk3 in promoting phosphorylation (Fig. 7e). Although myotube guidance has not been studied in vertebrate systems, Plk1 was recently shown to regulate myogenesis in mice. Muscle-specific deletion of *Plk1* blocked limb muscle development during embryogenesis, and prevented muscle stem cells from activating after injury[63]. It remains unclear if Vrk3 activates Plk1 during mammalian muscle development, if the targets of Plk1 in the muscle lineage have been conserved, or if Plk1 regulates myofiber morphogenesis. Our study highlights the exciting possibility that the role of Plk1 during vertebrate muscle development and regeneration extends beyond the mitotic events of myoblast proliferation and muscle stem cell activation.

## Methods

**Drosophila genetics**. The *bsd*[1] allele was recovered in an EMS screen as described[24]. The following stocks were obtained from the Bloomington Stock Center: *aurA*[1], *jar*[322], *lost*[1], *polo*[1], *polo*[KG03033], *tum*[DH15], *pav*[B200], *Df(2R)BSC199*, *Df(2R)BSC699*, *Df(3L)BSC447*, *Df(2L)Exel7049*, P{UAS-nod.GFP}, P{Gal4-tey*[5053A]*}, P{GMR40D04-GAL4}attP2 (*slou.Gal4*), P{GMR57C12-GAL4}attP2 (*nau.Gal4*), P{Gal4-how*[24B]*}, P{UASp-aurB.PrA}, P{UAS-eGFP}, P{UAS-Lifeact-RFP}, P{UAS-polo.T182D}, P{PTT-GC}*polo*[CC01326] (Polo-GFP), and the FUCCI lines P{UAS-GFP.nE2f1.1-230} and P{UAS-RFP.CycB.1-266}. The remaining stocks used in this study were *aurB*[1689] (see ref. [45]), P{Gal4-kirre*[rP298]*} and P{kirre*[rP298]*.nlacZ}[64], P{Gal4-Mef2}[65], and P{MHC.τGFP}[29]. *Cyo*, P{Gal4-Twi}, P{2X-UAS.eGFP}; *Cyo*, P{wg.lacZ}; *TM3*, P{Gal4-Twi}, P{2X-UAS.eGFP}; *and TM3*, P{ftz.lacZ} balancers were used to genotype embryos.

Bsd transgenic flies were generated by subcloning the *bsd* coding sequence (LD23371, *Drosophila* Genomics Resource Center, supported by NIH grant 2P40OD010949) into pUASt-Attb (KpnI/XbaI). Site-directed mutagenesis by PCR sewing was used to make UAS.Bsd.I129A. Plasmids and P(acman) BACs (CH321-61F090 and CH322-02P20) were injected and targeted to a φC31 integration site at 22A2 (Rainbow Transgenic Flies; Bloomington Stock 24481); stable insertions were identified by standard methods.

**RNA sequencing and variant identification**. Total RNA was collected from 12 to 24 h embryos per manufacturer's specification (RNeasy kit, Qiagen). cDNA libraries were generated with the TruSeq stranded mRNA sample library kit (Illumina) and sequenced using 50 bp paired-end reads on the Illumina HiSeq 2000 system. Two technical replicates of *w*[1118] and *bsd*[1] were prepared and

sequenced. Sequence reads were mapped to the *Drosophila* genome with Genomic Short-Read Nucleotide Alignment Program (GSNAP) using the Cufflinks method. Variants (single-nucleotide variants and insertions/deletions) were identified with the Broad Institute's Genome Analysis Toolkit (GATK), and the resulting variants were functionally tested by complementation test. The *bsd*[1] allele (Q545*) was confirmed by Sanger sequencing.

**Bsd antibody**. We created a fusion Bsd::6xHis fusion protein by PCR, using the C-terminus amino acids 705–1004 of Bsd. We subcloned the 598 bp fragment into the pHO4d 6xHIS expression vector[66] via conventional restriction enzyme sites. The Bsd 6xHIS fusion construct was transformed into competent BL21 (DE3) pLysS *E. coli* cells (Invitrogen) and grown, overnight shaking at 37 °C in DYT supplemented with 100 μg/ml ampicillin. The cells were diluted 25 times in fresh DYT media and grown at 37 °C to an OD$_{600}$ = 0.6–0.7. We added isopropyl β-D-thiogalactoside (IPTG) to 1 mM to induce expression of the fusion protein and incubated overnight shaking at 18 °C. We purified the 6xHIS fusion protein on nickel-nitrilotriacetic acid agarose (Qiagen, Valencia, CA) according to the manufacturer's protocols, under native conditions with modified buffers and dialyzed against PBS. We sent the purified protein to Pocono Rabbit Farm & Laboratory (Canadensis, PA) for guinea pig custom polyclonal antibody production.

**Plasmids and mutagenesis**. Expression plasmids for the immunoprecipitation screen were the BDGP Flag-HA C-terminal fusions FMO03130 (Lost), FMO06869 (Polo), FMO07294 (Imp), FMO11010 (Yp3), FMO12286 (Jar). Plasmids for expressing tagged proteins in S2 cells were generated by cloning coding sequences into pEntr/SD (Thermofisher, K242020), and recombining the coding sequences into pAc5 promoter destination vectors (pAWM and pAWF). Site-directed mutagenesis was performed as described above to generate Bsd.Q545* and Polo.T182A. To generate GST-Bsd for *E. coli* expression, the *bsd* coding sequence was subcloned into pGex4T-1 (Sal1/NotI). The Vrk3 mammalian expression construct was generated by recombining pDONR223-VRK3 (Addgene 23687[67]) into pDEST-CMV-3xFLAG-EGFP (Addgene 122845[68]); pRcCMV-Myc-Plk1 was used without modification (Addgene 41160[69]).

**Immunohistochemistry and in situ hybridization**. Antibodies used were α-Mef2 (1:1000, gift from R. Cripps), α-Tropomyosin (1:600, Abcam, MAC141), α-PLK1-phospho-T210 (1:100, Abcam, ab39068), α-GFP (1:600, Torrey Pines Biolabs, TP-401), α-GFP (1:300, Aves Labs, GFP-1020), α-dsRed (1:300, Takara, 632392), α-γ-tubulin (1:300, Sigma, T5326), and α-βgal (1:100, Promega, Z3781). Embryo antibody staining was performed as described[24]; HRP-conjugated secondary antibodies in conjunction with the TSA system (Molecular Probes) were used to detect primary antibodies. For S2 cell labeling, cells ($5 \times 10^6$) were transfected per manufacturer's specifications (Lipofectamine 3000, Invitrogen; applies to all transfections in this study), cultured at 25 °C in Schneider's *Drosophila* medium (Sigma, S9895) supplemented with 10% heat-inactivated fetal bovine serum (FBS, Invitrogen, 10082147) for 72 h, collected and washed once with S2 medium. Cells were then seeded into a 6-well-plate with a glass coverslip and incubated for 1 h. The cells were washed twice with PBS and fixed with 4% PFA for 15 min, and then washed three times with PBS. After 1 h blocking in 25%NGS/PBST, cells were incubated with α-FLAG antibody (1:1000, Sigma, F3165) and α-Myc antibody (1:1000, Sigma, PLA001) in PBST containing 0.5% BSA at 4 °C for 12 h. After five PBS washes, cells were mounted in Vectashield with DAPI (H-1000).

We performed in situ hybridization as previously described (Williams et al., 2015), except Dig-labeled probes were detected with HRP-conjugated α-Dig (Roche, 11207733910, 1:500) in conjunction with the TSA system (Molecular Probes). Probe templates were generated by cloning PCR products from DNA LD16949 and LD39409 into PCR2.1 vector. Templates were validated by Sanger sequencing.

**Imaging and image quantification**. Embryos were imaged with a Zeiss LSM800 confocal microscope; cells were imaged by confocal or with an inverted Zeiss

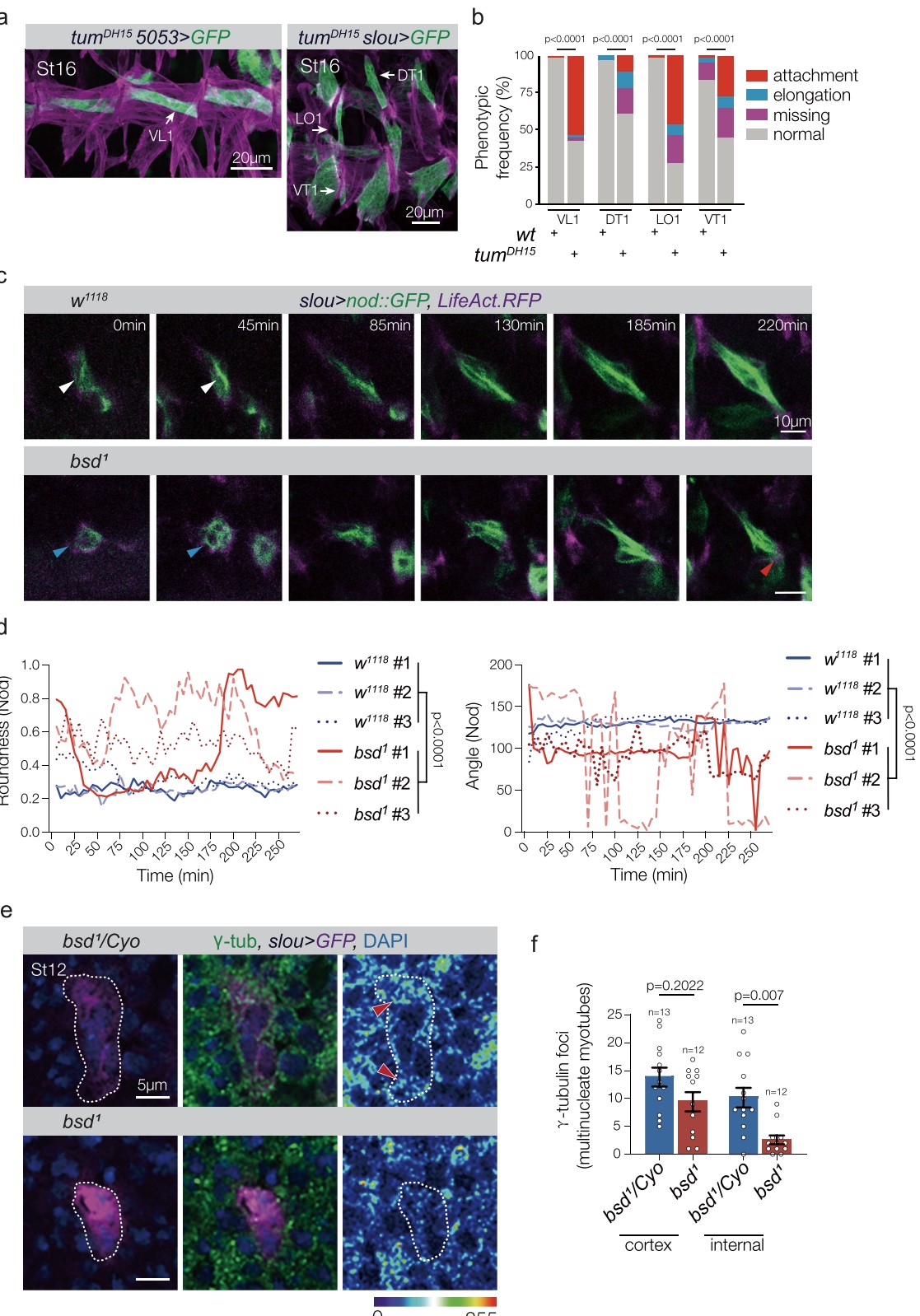

AxioObserver. For time-lapse imaging, dechorionated St12 embryos were mounted in halocarbon oil and scanned at 6 min intervals. For single-frame live imaging, embryos were dechorionated, mounted in PBT, and directly scanned. Control and mutant embryos were prepared and imaged in parallel where possible, and imaging parameters were maintained between genotypes. The fluorescent intensity and cell morphology measurements were made with ImageJ software.

**Phenotypic scoring, analysis, and visualization.** Each embryonic hemisegment has 30 distinct muscles with a fixed pattern as shown in Fig. 1b. Muscle phenotypes were analyzed in hemisegments A2–A7, in a minimum of nine embryos per genotype. For global muscle analyses, a Tropomyosin antibody was used to visualize all 30 muscles. For individual muscle analyses, GFP was expressed under the control of a muscle-restricted Gal4 driver to visualize a subset of muscles. For both assays, muscles

**Fig. 6 Bsd directs microtubule reorganization. a** $tum^{DH15}$ VL1, DT1, LO1, and VT1 muscle phenotypes. Stage-16 embryos labeled for *5053 > GFP* (green) or *slou > GFP* (green) and Tropomyosin (violet). Tum regulates microtubule reorganization during myogenesis[27], and $tum^{DH15}$ muscles showed attachment site defects similar to $bsd^1$ muscles. **b** Histogram of muscle phenotypes in *5053 > GFP* and *slou > GFP* embryos (wt, $n = 60$; $tum^{DH15}$, $n = 54$). **c** Live imaging stills showing LO1 myotubes in Stage-12–15 embryos that expressed Nod.GFP (microtubule minus ends, green) and LifeAct.RFP (F-actin, violet). Transgene expression was controlled by *slou.Gal4*. Live imaging initiated when Nod.GFP fluorescence was detectable (0 min). Control LO1 myotubes showed a linear array of microtubule minus ends at the onset of imaging (white arrowheads) that did not co-localize with F-actin. Nod.GFP remained cortical through the early stages of elongation in $bsd^1$ myotubes (blue arrowheads), and often overlapped with the F-actin domain (red arrowhead). See Supplementary Fig. 6D for F-actin heatmap. **d** Quantification of Nod.GFP distribution. GFP fluorescence was traced in each frame of three independent live-imaging experiments per genotype. Nod.GFP localization was stable in control myotubes, with a low roundness score (more linear). Myotube orientation (angle) was consistent throughout development. $bsd^1$ myotubes showed fluctuating Nod.GFP localization, with a high roundness score (more cortical), and incorrect orientation. $n = 162$ for each genotype. **e** Single confocal scans of multinucleate LO1 myotubes from Stage-12 *slou > GFP* embryos labeled for γ-tubulin (green), GFP (violet), and DAPI (blue). Control ($bsd^1$/Cyo) myotubes showed both cortical and internal cytoplasmic γ-tubulin foci, with internal foci concentrated toward the myotube leading edges (red arrowheads). $bsd^1$ myotubes had significantly fewer internal γ-tubulin foci compared to controls, but an equivalent number of cortical γ-tubulin foci. **f** Quantification of γ-tubulin foci in multinucleate Stage-12 myotubes. See Supplementary Fig. 6E, F for γ-tubulin foci in mononucleate founder cells. Embryos in (**e, f**) were derived from the same preparation. Significance was determined by two-sided Fisher's exact test (**b**), two-sided Wilcoxon signed-rank test (**d**) and one-way ANOVA (**f**). Error bars represent SEM.

were assigned a phenotype: normal, missing, elongation defect, attachment site defect. A "% Defective" was calculated for each of the 30 muscles in a minimum of 54 hemisegments. "% Defective = #" of abnormal muscles/hemisegments scored. % Defective was then converted to a schematic heatmap on the embryonic muscle pattern. "Histologic Score" was calculated using the following scale: missing = 3, elongation defect = 2, attachment site defect = 1, normal = 0. Histologic Score = sum of phenotypic score/number of embryos analyzed as described. Myoblast fusion was quantified by counting the number of lacZ+ myonuclei per hemisegment in *rP298.nlacZ* embryos. Fusion Index = #lacZ nuclei experimental/#lacZ nuclei control.

**GST pulldown and mass spectrometry.** GST-Bsd and GST were purified from *E.coli* by standard methods and stored at −80 °C in 1 ml aliquots. Twelve to twenty-four hours embryo lysates were collected by homogenizing dechorionated embryos in a Dounce homogenizer with 100 μl of lysis buffer (60 mM Tris pH 7.5, 80 mM NaCl, 6 mM EDTA pH 8.0, 2% Triton X-100, 5 mM 1-Naphthyl potassium phosphate, 2 mM PMSF, 1× Sigma Phosphatase Inhibitor II, 1× protease inhibitor) per 10 μl of embryos. The lysate was then centrifuged at 15,000 RCF for 10 min to pellet large debris. The supernatant was diluted to a final protein concentration of 1 mg/μl, aliquoted in 100 μl volumes, and flash frozen. For affinity purification, 500 μl of dialyzed GST-Bsd or GST was bound to 50 μl of PBS washed glutathione sepharose beads (GE Healthcare, 17-0756-01) and incubated at 4 °C for 30 min. The beads were washed with PBS-1% Triton X-100. Embryo lysates (100 mg protein/ml) were incubated with the protein-bound beads at 4 °C for 4 h; the beads were then washed three times and submitted to the Washington University Proteomics Core Lab for liquid chromatography-mass spectrometry (1260 Infinity II Bio-Inert LC System, Agilent Technologies).

**In vitro Bsd kinase assay.** For cell-free in vitro kinase assay, GST-Bsd and GST-Polo were purified from *E. coli* as described above. GST-Bsd were incubated with GST-Polo for 30 min at 30 °C in 20 mM HEPES (pH 7.5), 2 mM MgCl₂, 1 mM DTT, and 0.5 mM ATP. Reactions were stopped by the addition of 5× SDS loading buffer. Samples were separated by SDS-PAGE and blotted with α-PLK1-phospho-T210 (1:500, Abcam, ab39068) and α-GST (1:1000, Cell Signaling Technology, #2625).

**Immunoprecipitation and western blotting.** For *Drosophila* proteins, S2 cells ($8 \times 10^6$) were transfected with 2 μg of each plasmid in six-well plates. Cells were cultured for 48 h, incubated with 2 mM CuSO₄ for 24 h (for FMO plasmids only), collected, washed twice with PBS, lysed with 600 μl IP buffer (20 mM Hepes, pH = 7.4, 150 nM NaCl, 1% NP40, 1.5 mM MgCl₂, 2 mM EGTA, 10 mM NaF, 1 mM Na₃VO₄, 1× proteinase inhibitor), incubated on ice for 30 min, centrifuged at 12,000×g for 15 min. The supernatant was collected, incubated with 2 μl α-FLAG (Sigma, F3165) or α-Myc (PLA001, Sigma) overnight at 4 °C, and then incubated with 30 μl Dynabeads (Invitrogen, 10007D) for 4 h at 4 °C. The beads were washed 5× with IP buffer, and immunoblotted with α-Myc (1:3000) or α-FLAG (1:2000)[70].

For in vitro phosphorylation assays, immunoprecipitation was carried out as described, except that anti-Phosphothreonine antibody (1:125, Abcam, ab9337) was used for immunoblotting. For in vivo phosphorylation assays, 200 *Polo-GFP* and 200 $bsd^1$; *Polo-GFP* embryos were homogenized in 600 μl IP buffer, large debris was removed by 15 min centrifugation (12,000×g), and immunoprecipitation was carried out as described above using 2 μl α-GFP (Torrey Pines Biolabs, TP-401).

For mammalian proteins, HEK293T cells were seeded in 6-well plates, grown to 60% confluency at 37 °C and 5% CO₂ in Dulbecco's modified Eagle's medium (DMEM; Invitrogen) supplemented with 10% heat-inactivated FBS. Cells were transfected with 2 μg of each plasmid and cultured for 48 h. Immunoprecipitations

were carried out as described above. Plk1 phosphorylation was directly assayed without immunoprecipitation using α-PLK1-phospho-T210 (1:500, Abcam, ab39068).

Western blots were performed by a standard method using precast gels (#456-1096, BioRad), and imaged with the ChemiDoc XRS + system (BioRad).

**siRNA knockdown and inhibitor treatments.** For siRNAs, C2C12 cells were seeded in six-well-plate and grown in standard conditions to 60% confluency in growth medium (10% FBS in DMEM), and transfected 10 nM duplexed 27nt siRNAs (Integrated DNA Technologies). Transfection efficiency was monitored with Cy3 transfection controls (Trifecta Kit, Integrated DNA Technologies). After 24 h, the growth media was changed to differentiation media (2% horse serum in DMEM). After 7 days of differentiation, cells were fixed for 15 min in 4% PFA and stained with α-alpha-actinin antibody (A7811, Sigma, 1:1000).

For Volasertib, C2C12 cells were seeded in a six-well plate and grown in standard conditions to nearly 100% confluency in growth medium, and treated with 100 nM Volasertib or DMSO for 24 h in growth medium (No.S2235, Selleck chem). Then the growth medium was changed to differentiation medium (2% horse serum in DMEM), with 100 nM Volasertib or DMSO, and incubated for 48 h. Cells were incubated in differentiation medium without Volasertib or DMSO for additional 5 days, fixed and stained as described above.

**Quantitative real-time PCR.** Total RNA was extracted with RNeasy mini kit (74104, Qiagen), and quantified (Nanodrop 2000). The cDNA was prepared by reverse transcription with M-MLV Reverse Transcriptase (28025013, Thermo) with 2000 ng RNA. PowerUp Sybr Green Master Mix (A25742, Thermo) and ABI StepOne system (Applied Biosystems) were used for quantitative RT-PCR. Quantification was normalized to *GAPDH* or *RpL32*. Primers used were:
Vrk1-F-5′-ACAGGTTTATGATAATGGACCGC-3′
Vrk1-R-5′-CTGGTCAGGGTTCTTGTGACT-3′
Vrk2-F-5′-CCGCACATGGACACTCTGTA-3′
Vrk2-R-5′-CTTGCTGGATGAACTCCCAG-3′
Vrk3-F-5′-ATCAAGGACCCAGAAGTGGAGA-3′
Vrk3-R-5′-TTCTTCCATTTGTTCACTTGCAGA-3′
Gapdh-F-5′-TGTAGACCATGTAGTTGAGGTCA-3′
Gapdh-R-5′-AGGTCGGTGTGAACGGATTTG-3′
Bsd- F-5′-TCAACGCTAAGCACTCCGTT-3′
Bsd-R-5′-CGCCTCTGCTCCATGTCTAG-3′
Rp32- F-5′-ATGCTAAGCTGTCGCACAAATG-3′
Rp32- R-5′-GTTCGATCCGATACCGATGT-3′

**Bioinformatic and statistical analysis.** Protein alignments were generated in ClustalX, and phylogenetic analyses were performed with DNAMAN (Lynnon Corporation) using the observed divergency distance method. ATPbind was used to predict Bsd ATP-binding residues; informatics predictions were compared to the VRK1 ATP-binding pocket described in ref. [33].

Statistical analyses were performed with GraphPad Prism 8 software, and significance was determined with the unpaired, two-sided Student's *t* test, one-way ANOVA, two-way ANOVA, two-sided Fisher's exact test, two-sided Wilcoxon signed-rank test or Mann–Whitney test (for non-Gaussian distributions). Gaussian distribution fit curve and skew distribution fit curve were generated with Origin 2019 software. Sample sizes are indicated in the figure legends. Data collection and data analysis were routinely performed by different authors to prevent potential bias. All individuals were included in the data analysis.

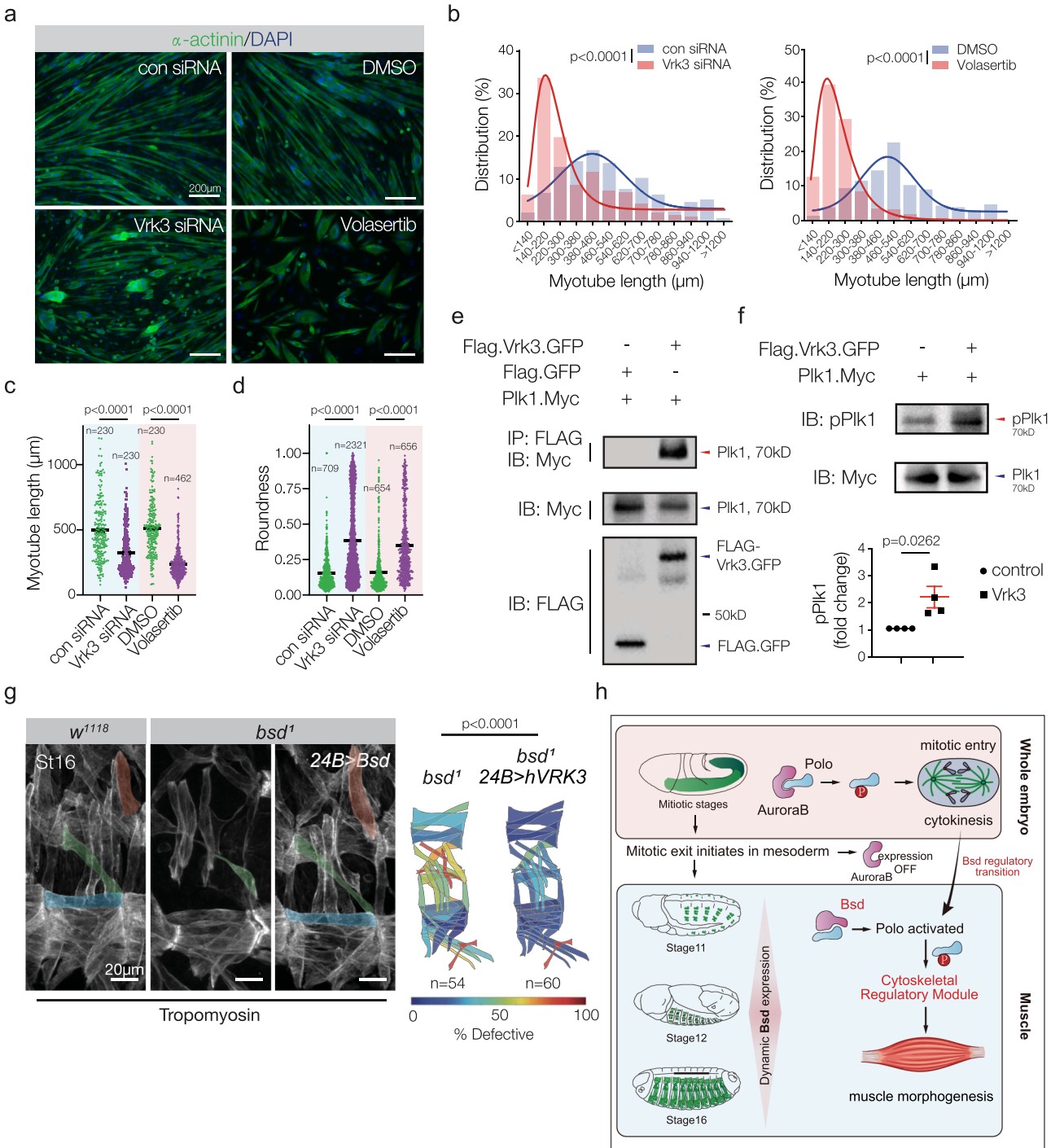

**Fig. 7 The Bsd orthologue Vrk3 is required for myotube elongation. a** C2C12 cells treated with siRNAs against murine Vrk3 or with an inhibitor of Plk1 (Volasterib, 100 nM). Cells were fixed after 7 days in differentiation media, and labeled for α-actinin (green) to detect differentiated myotubes. Vrk3 knockdown and Volasterib treated myotubes were shorter than controls and were often rounded. **b** Myotube length distribution. Solid lines show the Gaussian distribution fit curve (blue) and skew distribution fit curve (red). Vrk3 RNAi and Volasterib treated myotubes were shorter than controls. For each group, n = 230. **c** Quantification of cumulative myotube length. **d** Myotube roundness scores. Vrk3 RNAi and Volasterib treated myotubes showed a higher roundness score, indicating increased circularity. **e** Immunoprecipitation of HEK293 cell lysates showed a physical interaction between Vrk3 and Plk1. **f** Western blot of HEK293 cell lysates transfected with Vrk3 and Plk1. Vrk3 promoted Plk1 phosphorylation at Thr210. For each group, n = 3. **g** Control (w[1118]), bsd[1] UAS.VRK3, and bsd[1] 24B.Gal4 > VRK3 embryos labeled for Tropomyosin. DT1, LO1, and VL1 muscles are pseudocolored orange, green, and violet. bsd[1] embryos that expressed human VRK3 in the mesoderm showed improved muscle morphology (**h**) Model showing Polo activation transitions from AurB in mitotic tissues to Bsd in post-mitotic tissues. Polo controls cytoskeletal regulatory module to direct cytokinesis and myotube guidance. Significance was determined by unpaired Student's t test (**b, d**), Mann–Whitney U test (**c**) and Fisher's exact test (**g**). Error bars represent SEM.

**Source information for reagents**. Information for all reagents used in this study is reported in Supplementary Data 3.

**Reporting summary**. Further information on research design is available in the Nature Research Reporting Summary linked to this article.

## Data availability

The data that support all experimental findings of this study are available in multiple formats. Data necessary to reproduce all statistical analyses and results in the paper are provided in the Source Data File provided with this paper, and raw data are available at https://doi.org/10.17605/OSF.IO/YX7CR. RNA-seq data analyzed in this manuscript has been previously published and is available at https://doi.org/10.5061/dryad.j0zpc869m. Mass spectrometry data are available on the PRIDE database under accession number PXD030953. Source data are provided with this paper.

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

## Acknowledgements
We thank Helen McNeill and Mayssa Mokalled for critical reading of the manuscript, Jean-Rene Huynh and the *Drosophila* community for stocks and reagents. Our LC-MS was performed by the Washington University Proteomics Shared Resources, supported by NIH UL1 TR000448. A.N.J. was supported by NIH R01AR070299. P.H.T. was supported by NIH R01MH067122.

## Author contributions
Conceptualization: A.N.J. and S.Y.; methodology: A.N.J.; formal analysis: S.Y., J.MA., Y.D., J.T., P.H. and A.N.J.; investigation: S.Y., J.MA., Y.D., J.T., P.H and A.N.J.; resources: P.H. and A.N.J.; data curation: S.Y., Y.D. and A.N.J.; writing original draft: A.N.J. and S.Y.; visualization: S.Y. and A.N.J.; supervision: S.Y. and A.N.J.; project administration: A.N.J.; funding acquisition: A.N.J.

## Competing interests
The authors declare no competing interests.
