## [Peer Review File · Nature Communications]

Spatiotemporal expression of regulatory kinases directs the transition from mitosis to cellular morphogenesisReviewers' comments:

Reviewer #1 (Remarks to the Author):

In this manuscript, Yan et al report that the uncharacterized kinase that they name Back seat driver (Bsd) is required for muscle development in *Drosophila*. In searching for protein interactors of Bsd, they find the Polo kinase. The authors then proceed to study the functional relationship between the two kinases. Based on their observations, they propose that Bsd phosphorylates Polo at its major activation site Thr182, and that activated Polo in turn promotes cytoskeletal rearrangements required in myoblasts.

The identification of Bsd and its requirement in muscle development is exciting and some of the subsequent observations linking it with Polo are enticing and pointing in the direction proposed by the authors. However, their results often fall short of supporting the conclusions put forward. Therefore, this work is too preliminary for publication and more experiments are needed. The presentation of the results is deficient and crucial information is lacking in many places.

MAJOR COMMENTS

The biochemical experiments are insufficient and some of them are flawed or their results are poorly presented. Here are some problems:

A major piece missing from this study is the demonstration that Bsd directly phosphorylates Polo at Thr182. The authors should purify Bsd and test if it phosphorylates purified Polo (preferably kinase-dead) *in vitro*. Polo-T182A should be used as a control substrate to verify that the detected phosphorylation occurs at Thr182. They should also use the anti-pThr210-PLK1 (Abcam, ab39068) to detect pThr182-Polo by Western. This antibody was used successfully to detect pThr182-Polo in several previous publications (see for example Carmena et al, 2012, PLoS Biol).

In addition, the authors should assess the Thr182 phosphorylation of Polo *in vivo* (flies or cells) using the anti-pThr210-PLK1 antibody instead of a generic antibody against pThr (Fig 4A, C, S4B). The problem is that Polo can be phosphorylated at several Threonine residues that may or may not contribute to its activation.

In Figure 3A's legend, it says proteins were "purified with control or Bsd-bound GST beads". This phrasing is unclear. From reading the M&M section, I conclude that they purified from bacteria GST-Bsd or GST on glutathione Sepharose that they then used to pulldown interacting proteins from embryo extracts. In the figure, why do they write "Bsd.GST", and why not indicate the GST control? Unless the control was something else? Also, the authors simply provide a color for each bait protein following a color code pretended to reflect abundance. This is insufficient. At a minimum, they should provide at a supplementary table with the peptides identified and the total spectral counts for each protein.

Fig S3A makes little sense. If the control is GST alone, it should be indicated. What is shown in lanes 2 and 3? In both, we see a strong band at 25 kDa, the MW of GST. Where is GST-Bsd? I would expect a band at a much higher MW. And if lanes 4 and 5 are like 2 and 3 but with the addition of embryo lysate, then why do we not see the 25 kDa band like in lanes 2 and 3? Is the elution different between 4 and 5 vs 2 and 3?

The data presented in Fig 3B is inconclusive. Bsd-Myc and Bsd.Q545*-Myc should be shown on the same membrane to compare levels (both from the extracts and the IP products). A Western of the Polo-Flag IPed should also be shown. The same requirements apply to results shown in Fig 3C and 6E.

Fig 4A and C: The total IPed protein (Polo-Flag or Polo-GFP) should be blotted to compare

phosphorylated / total.

Fig 4B: PoloT182A + Bsd should be tested in parallel and results shown.

Fig 4D: Which antibody was used to stain pPolo? This is nowhere indicated. I hope it's not the generic pThr antibody.

Movie 2 needs a WT control for comparison.

Figure S4B does not show an in vitro phosphorylation assay as indicated in the legend. It is simply a Western from an IP.

The last part of the manuscript attempts to connect Polo with Tumbleweed and Pavarotti as effectors in the regulation of the cytoskeleton during myotube formation. Unfortunately, the evidence presented is far from sufficient in many ways. First, there is no previous evidence that Tumbleweed or Pavarotti is a phosphorylation target of Polo regulated in their function by Polo in *Drosophila*. This manuscript does not provide any such evidence either. Suppositions are made based on what is known about mammalian orthologs in cytokinesis. But they would need to be tested here in myoblasts in *Drosophila*. The implication of Tumbleweed in myotube formation is limited to the data presented in Fig 4G, H. No biochemical or genetic link is established with Polo and nothing at all is shown about Pavarotti. Also, I am not convinced that the foci seen with the g-Tub antibody in Fig 5D and S5 are specific or meaningful in any way. They simply seem to be everywhere in the cytoplasm.

No statistics are done to analyzed the data presented in Figs 1B, F, G, 2B, 3F, G, 4F, H, 5C, 6C, S3B, C. It seems that experiments were done only once.

I don't understand the QQ plots shown in Fig S6C. What are the values being compared? What are the units on the axes?

The model shown in Fig S6D is extremely premature.

Lines 190-192: It is written: "The Polo/Tum/Pav cytoskeletal regulatory module interacts with the microtubule cytoskeleton in post-mitotic myotubes, suggesting microtubules are the major target of this cytoskeletal regulatory module (Fig. S6D)." What interaction are the authors referring to? No such data is shown in the manuscript. Is this statement referring to a previous paper? In this case, there would be a reference missing?

MINOR COMMENTS

Line 25. I'm not sure what "mitotic growth" means. Mitosis and growth are separate processes. I suggest replacing with "mitosis".

Lines 95-96: The term "Affinity Purification and Mass Spectrometry (AP-MS)" is used incorrectly as it always refers to the purification of a tagged protein and the identification of the proteins co-purified. Here, the authors are doing GST pulldowns and they should call it that way.

Lines 137-8: The increase in pHH3 cells upon expression of AurB is interpreted as an increase in mitotic index. However, as Ser10 of pHH3 is a direct target of AurB, the increased phosphorylation does not necessarily reflect a mitotic state. Conclusions should be adjusted accordingly.

Fig S4D: The distance units should be indicated.

Lines 221-223: I don't understand the logic behind this statement: "In fact, the conservation of Bsd/Vrk3 cellular and molecular functions is so striking that Vrk3 likely regulates Plk1 activity under a

variety of developmental and homeostatic contexts.”

Reviewer #2 (Remarks to the Author):

Key results

This manuscript describes an extensive piece of research that embraces an impressive array of experiments/assays and there is a lot of very good content woven into a satisfying narrative. The study goes from the outcome of a genetic screen that identified a gene that affects muscle development through to analysing aspects of the gene's (bsd) mechanism of action that indicate a previously undescribed role for a mitotic kinase (Polo) in aspects of muscle cell differentiation. First, the (relatively mild) muscle phenotype of the uncharacterised kinase Bsd is analysed. Then protein interaction assays (Mass Spec and IPs) identify the mitotic kinase Polo as a Bsd-interacting protein. A muscle phenotype for polo is shown, and bsd promotes Polo phosphorylation in embryo lysates. Strikingly, it is demonstrated that expressing activated Polo in the developing mesoderm can rescue the bsd mutant muscle phenotype, implying that Polo is a critical downstream effector of Bsd action in muscle cell differentiation. Evidence is provided for a mechanism by which Bsd can affect muscle cell guidance through regulating microtubule dynamics. Lastly, results and discussion are presented that suggest conservation of the mechanism in mammalian muscle cells. The novel suggestion presented here is that a mitotic kinase (Polo) with a documented function in the control of cell division also has a role in post-mitotic cells in aspects of cell differentiation (“cell guidance” in the authors’ turn of phrase). The question is then how does Polo affect cell divisions and subsequently cell differentiation? From the title onwards, the authors emphasise the spatial/temporal expression aspect of the expression of the Polo regulators Bsd and Aurora B in order to explain. In my view (see below), aspects of this angle (and some others) would benefit from improved clarity and precision.

Originality/Significance

The transition from dividing to overtly differentiating muscle cells has been studied in culture for many years. However, this transition is much less studied in vivo in developing animals, and so the insight provided by this study is very welcome. This work highlights the microtubules of the cytoskeleton. There are very different cytoskeleton behaviours in cell division and cell guidance. One might anticipate that possibly some similar players might be involved in the co-ordination of these different processes, although I am not aware of a previous specific implication, as made in this study, of a specific player like Polo.

This study therefore has a core content of real interest (and not just for muscle biologists), but in general I feel that more detail and clarity on aspects of the methodology and the presentation of the results obtained would be helpful for the reader and would further improve the m/s. A key point is to be more precise about what is happening when and where in terms of expression of Bsd, Polo and AuroraB, in relation to cell division and cell morphogenesis.

Some examples follow.

Data and Methodology

Bsd expression

One important point is how clear is the expression data to support the statement in the Introduction (157): “the transition from mitotic growth to cellular morphogenesis is achieved through the spatially and temporally restricted expression of the Aurora kinases and Bsd”.

Start with Bsd.

L79 “Bsd is ubiquitously expressed in blastoderm embryos, but after gastrulation Bsd expression in the mitotic mesoderm was significantly reduced”. It could be, but no measurements are presented, and the embryo images are small and mesoderm is not labelled making assessment by the reader difficult, especially for the non-specialist.

L81 “Bsd expression became progressively enriched”. It is difficult for the reader to see this from what

is presented. The plot in a supplemental figure (Fig S2 E) is the clearest to show an increase in nuclear Bsd between stages 11 and 14, but even this is dependent on a constant level in the non-mesoderm nuclei (is this known to be the case?). The authors should explain the evidence that lies behind using this ratio of intensities.

The main Fig 2A is not compelling as support. Stage 14 expression looks less than stage 12, but Fig 2SE says it is higher.

The Fig 2 legend also refers to nuclear/cytoplasmic distribution. This is not clearly shown, not least taking into account the different distribution of nuclei in the myogenic mesoderm at stage 11, compared to stages 12 and 14.

Bsd muscle phenotype

In Fig 1 the authors have taken a thorough approach to the challenge of analysing and presenting a relatively mild (and variably penetrant) muscle phenotype. However, the muscle phenotype scoring, especially for readers not familiar with the *Drosophila* embryo musculature, could be more completely explained. Maybe because of this, I had difficulty in relating the heat map of Fig1B to the plot in Fig1G.

Fig 1 legend says "individual muscles were scored in segments A2-A8 of st 16 embryos". Each abdominal segment A2-A8 contains the same stereotypic muscle pattern and there are two of each muscle (e.g. muscle VL1) in each segment, one on each side of the midline, i.e. 14 examples of each muscle in A2-A8 per embryo. However, if the authors actually mean the more usual hemi-segments, then this number reduces to 7 per embryo.

The plot in Fig 1G says it shows the sum of phenotypic scores divided by number of embryos. To readily interpret this, it should say how many muscles were scored per embryo. It would be 7 if 7 hemi-segments were scored, 14 if 7 segments. Thus, a "histologic score" of 7, e.g. for muscle VT1, would mean an average score of 1 per muscle (if 7 muscles per embryo).

The score of "3" for VL1 would mean an average score of 0.43 per muscle. In the scoring system used, muscle VL1 would thus not be missing, very rarely would have its elongation affected, and a little less than half of them would have attachment affected. However, the shade of blue on the heat map in Fig 1B suggests VT1 is only affected in 20-25% of examples, so maybe the number of muscles per embryo the authors are using is 14, i.e. they count segments.

Which step of muscle differentiation does *bsd* affect?

In order to conclude that *bsd* affects muscle cell guidance, an important point is that it doesn't affect a step in the muscle development pathway prior to this. The authors addressed this by asking about muscle precursors/founders. However, only a supplemental figure (Fig S1E,F) is presented to support the statement (l71) that "Bsd is not required for muscle precursor specification". The images are not the highest quality, but they are quantified and this shows that the number of muscle precursors is unchanged. It would help the reader, especially the non-specialist, if they used the same terminology. Thus, they refer to muscle "precursors" in the text (l72), but to "founder cells" in the figure legend. It would also help if the use of 5053>GFP and *slou*>GFP was explained. These markers together only address 4 of the 30 founder cells per hemi-segment. Moreover, they don't mark any of the founders for the five muscles that are most affected in the *bsd* mutant in Fig 1B (LT4, DO3, DO5, VA3, DA3). A marker for all founders (e.g. *rp298*) could be used.

Polo

It would be helpful to include more information on the *polo* hypomorph allele. Does it have a phenotype prior to muscle cell guidance? One would expect many *polo* mutants to have earlier phenotypes that might influence how muscle develops, e.g. affect an aspect of cell division that could affect the myoblast pool available for muscle development. As for *bsd*, the question also arises as to whether *polo* affects muscle precursor/founder specification. A gap in the manuscript is that only a stage 16 (near the end of embryogenesis) muscle phenotype is shown.

Mesoderm mitotic cell divisions

When considering the role of Polo plus AurB in cell division, and Polo plus Bsd in muscle cell guidance,

important aspects are when are mitotic cell divisions complete in the myogenic mesoderm, and what is the expression of these three players with respect to this time.

The Model in Fig 6G indicates mitosis is complete in the mesoderm by before stage 11. However, although this is the case for the third mesodermal cell division, the fourth mesodermal division continues in stage 11 and 12. (Bate (1993), *The mesoderm and its derivative*, pp1013-1090, in *The Development of Drosophila melanogaster*, CSHL Press). Indeed, FigS4E shows myogenic mesoderm cell division still continues in stage 11.

L135 "a majority of cells in the mesoderm exit the cell cycle prior to the onset of myotube guidance". This implies that the transition between cell division and morphogenesis in the whole population is not uniform, and there is need to assay on a cell-by-cell basis in order to establish the expression of these players with respect to the change from mitosis to muscle cell differentiation.

Aurora expression

L115 – "since AurB and AurA are not expressed in post-mitotic mesoderm during muscle morphogenesis". This is important for the authors' hypothesis for how the control of Polo kinase activation changes from AurB during mesodermal mitosis to Bsd during muscle cell guidance. The support for this idea needs more precision. When is the last mitosis in the myogenic mesoderm? (see above) What is the evidence that there is no mesodermal Aurora expression after this point? The authors just cite refs 44 and 45 in support. Both are in situ hybridisations (not protein) with little detail. 45 is the bulk assay of gene expression in the Drosophila genome project. These citations do not seem sufficient for this important point. This really needs the authors to include their own high quality study specifically addressing the question of the timing of any Aur B (and A) expression in the mesoderm.

Other minor suggested improvements

- In the Introduction (l52) it would help the reader to know here when mitosis finishes in the developing mesoderm, before (s)he considers when different players in this study are expressed.
- l73 states "Bsd directs myotube elongation and muscle attachment site selection". This conclusion would be strengthened by using a marker for attachment sites. The question is: have the attachment sites changed position, or do the developing muscles miss their usual site and attach elsewhere? If the attachment site positions change, this questions the mesoderm cell autonomous function of bsd.
- l92 (and Fig 2): is the Bsd.l129A mutant expressed (at the same level as the WT)? This needs to be shown in order to conclude it doesn't have the effect of the WT.
- l100 says "the truncated Bsd.Q545* protein". This would be useful to include in paragraph 1 of the results.
- Fig 1D: wt and bsd1 are labelled the wrong way around.
- Fig 1F: the histogram is not labelled to indicate which muscle the results are from.
- From the Mass spec results (Fig 3A), explain why Polo was pursued (it is low down on the presented list).
- l122: what stage are the embryos in the experiment that shows less in vivo Polo phosphorylation in bsd mutants.
- Fig5: explain what LifeAct.RFP is.
- References: A sentence in the Introduction (l48) as written seems to imply that the references 28-31 support the comment that "muscles in Drosophila are easily visualised in live, unperturbed embryos". These references are not for live embryos.
- Methods: The authors say in three sections "include" when listing reagents used (l233, l267, l278). This gives the impression that not all reagents are listed.

Conclusion/Interpretation

Overall, it didn't feel that the experiments undertaken and the results obtained were always matched by the thoroughness of the explanations and descriptions. This should be readily addressable. The manuscript needs to be tightened up in some of the conclusions the authors emphasise (see above).

Clarity and context

Title: Do the expression results presented support the title?

Abstract (I11) says "The transition between mitotic growth and morphogenesis is accomplished through the spatiotemporal transcriptional regulation of AurB and Bsd". However, "transcriptional regulation" has not been shown. In the introduction (I59) the authors just say "restricted expression". Even described like this, how clearly has this been shown? (see other comments).

The first paragraph of the Discussion is a succinct, clear, accurate summary. Some of the other sections could follow a similar presentation.

Reviewer #3 (Remarks to the Author):

This manuscript by Yang, et al., identifies a role for the previously uncharacterized *Drosophila* kinase CG8878 (named *bsd* by the authors) in directing post-mitotic functions of myogenesis, specifically myotube guidance and attachment site selection. MS experiments reveal binding to Polo, a kinase well studied during mitosis. The authors hypothesize that AurA phosphorylates and activates Polo during early mitosis, while Bsd assumes this role after mitosis is complete. The paper is clearly written with beautifully presented data. Overall, the work is novel as there are few post-mitotic roles for Polo and the paper is of interest in the field of muscle development. Some of the statements overstate the data and do not fully support the hypothesis. More evidence is necessary to conclude that *bsd* alone is responsible for Polo activation. Major comments that must be addressed are below.

Major comments

(1) Table S1 is missing.

(2) There does not appear to be much difference in Bsd mesoderm staining from stages 13 to 16 to support the claim that 'but during the final stages of myogenesis Bsd expression in the mesoderm was again reduced.' (p. 4, lines 82-83). Even in Figure 2A the expression of Bsd seems present in stage 16, just in the cytoplasm (and possibly slightly enriched in nuclei at muscle ends).

(3) Are *bsd* mutants embryonic lethal? Does loss of *bsd* also affect myoblast fusion? Panels 1E, 2B, 4F, 4G and possibly panels S1B (small pictures) show unfused myoblasts. siRNA of Vrk3 (panel 6A) also shows fusion defects. If true, these data suggest that Bsd and Vrk3 are required for an additional post-mitotic function. This should be clarified.

(4) Does *ballchen* have embryonic muscle morphogenic defects? There are reported muscle defects for *ballchen* RNAi knockdown later in development, so could it be partially redundant with CG8878? Note that many muscle looks OK in *bsd* mutants.

(5) In the AP-MS experiments, were other candidate protein interactions verified? Via the heat map, Polo was not as strong as some other candidates.

(6) Is it conclusive that AurA and AurB are not expressed in mesoderm tissue? Even if RNA in situ data shows decreased levels in mesoderm (per Flybase), there may still be some faint signal. The Flybase Developmental proteome data also shows relatively high protein expression for AurA and AurB after stage 12 (i.e., is this nervous system specific or is there some protein expression in muscle tissue)? This needs to be established as a major tenant of this work is the idea that AurA switches it Polo activation role to Bsd.

(7) What are explanations for the lack of complete rescue of *bsd* mutants by expression of PoloT182D? Does expression of PoloT182D on its own give a phenotype?

(8) The experiment showing that misexpression of AurA by *mef2-Gal4* prolongs mitosis is intriguing, but does not prove that activation of Polo switches from AurA to Bsd. What if AurA is indeed expressed, but held inactive and/or regulated, in muscle tissue under normal conditions. Overexpression of AurA may simply overcome this inhibition. What if activated Polo is expressed in AurA and/or AurB backgrounds?

(9) Both Tum and Pav have been shown to be Polo effectors. Only Tum is examined in this manuscript. Thus, some sentences need to be modified to focus only on Tum (p., 7, lines 149-150; p. 8, line 163; 9, line 190).

(10) How exactly is Polo directing MT organization? Multiple reports show that Polo directs MT

nucleation. This does not appear to be discussed or specifically tested (other than showing that gamma-tubulin is aberrant, which could be direct or indirect).

(11) Can Vrk3 substitute for Bsd to rescue muscle morphogenesis defects?

(12) The statement on p. 8, line 162 is not quite accurate. Bsd is not an essential regulator of MY dynamics. Features of dynamics (treadmilling, etc) were not tested.

(13) Guidance implies that factors influence direction and attachment site specificity. How does the Bsd Δ Polo pathway fit into this paradigm? This must be discussed.

Minor comments

(1) The phrase 'in turn to the skeleton' doesn't read well. There also is needs to be a bit more in the introduction about the conservation between Drosophila and mammalian myotube guidance (p. 3, line 48). What is known? What remains to be uncovered?

(2) Figure S1A – I believe 'Not data' should be 'No data available' underneath the graphs

(3) p. 5, lines 110-111. Is this statement referring to Drosophila or mammalian proteins?

(4) Please double

Although we cannot offer to publish your paper in Nature Communications, the work may be appropriate for another journal in the Nature Research portfolio. If you wish to explore suitable journals and transfer your manuscript to a journal of your choice, please use our manuscript transfer portal. If you transfer to Nature-branded journals or to the Communications journals, you will not have to re-supply manuscript metadata and files. This link can only be used once and remains active until used.

All Nature Research journals are editorially independent, and the decision to consider your manuscript will be taken by their own editorial staff. For more information, please see our manuscript transfer FAQ page. Note that any decision to opt in to In Review at the original journal is not sent to the receiving journal on transfer. You can opt in to In Review at receiving journals that support this service by choosing to modify your manuscript on transfer. In Review is available for primary research manuscript types only.

Reviewer #1

Comment: The identification of Bsd and its requirement in muscle development is exciting and some of the subsequent observations linking it with Polo are enticing and pointing in the direction proposed by the authors.

Response: Thank you for the enthusiasm!!!!

Comment: A major piece missing from this study is the demonstration that Bsd directly phosphorylates Polo at Thr182. The authors should purify Bsd and test if it phosphorylates purified Polo (preferably kinase-dead) *in vitro*. Polo-T182A should be used as a control substrate to verify that the detected phosphorylation occurs at Thr182.

Response: This is a good point. We attempted to GST purify Polo multiple times prior to submitting the original manuscript, but we simply could not recover Polo from *E. coli* (in PMID: 27093086 the authors purchased PLK1 protein for kinase assays even though the paper is focused on Polo-Miro interactions; Polo purification appears to be generally problematic). Our approach of precipitating Polo from cells and tissues, and then assaying phosphorylation is an acceptable method to show the phosphorylation of one protein is dependent on a second protein (eg PMID: 27093086, PMID: 20534669). In total, we show Polo phosphorylation is Bsd dependent with 5 independent experiments:

1. Polo IP'd from cells transfected with Bsd, detected with pThr (Fig. 4A).
2. Western Blot of phosphor-Polo from cells transfected with Bsd, detected with pThr210-PLK1 (Fig. S4C).
3. Polo IP'd from *bsd* mutant embryos, detected with pThr (Fig. 4C).
4. pPolo immunolabeling in S2 cells transfected with Bsd, detected with pThr210-PLK1 (Fig. S4D).
5. Phospho-Polo in the embryonic mesoderm of *bsd* mutant embryos, detected with pThr210-PLK1 (Figs. 4D, S4E).

We deliberately avoided using the term 'direct' when discussing Polo phosphorylation in both the original and the revised manuscript.

Comment: They should also use the anti-pThr210-PLK1 (Abcam, ab39068) to detect pThr182-Polo by Western. This antibody was used successfully to detect pThr182-Polo in several previous publications (see for example Carmena et al, 2012, PLoS Biol).

Response: We had shown Polo phosphorylation is Bsd dependent *in vivo* using anti-pThr210-PLK1 in the original manuscript (Fig. 4D), and now provide the Western Blot (Fig. S4C), and immunohistochemistry in S2 cells (Fig. S4D) in the revised manuscript.

Comment: In addition, the authors should assess the Thr182 phosphorylation of Polo *in vivo* (flies or cells) using the anti-pThr210-PLK1 antibody instead of a generic antibody against pThr (Fig 4A, C, S4B). The problem is that Polo can be phosphorylated at several Threonine residues that may or may not contribute to its activation.

Response: We completely agree!!! We had shown Polo phosphorylation is Bsd dependent in vivo using anti-pThr210-PLK1 in the original manuscript (Fig. 4D). The broad (generic) pThr antibody was necessary to show refinement toward p182. The pThr experiments suggested that Bsd broadly promotes Polo phosphorylation. Once we established the relationship between Bsd and Polo, then we could ask if Bsd promotes phosphorylation at Thr182. Although we had provided the in vivo results previously, we have added a Western Blot of pThr210-PLK1 in Bsd transfected cells in the revised manuscript (Fig. S4C).

Comment: In Figure 3A's legend, it says proteins were "purified with control or Bsd-bound GST beads". This phrasing is unclear. From reading the M&M section, I conclude that they purified from bacteria GST-Bsd or GST on glutathione Sepharose that they then used to pulldown interacting proteins from embryo extracts. In the figure, why do they write "Bsd.GST", and why not indicate the GST control? Unless the control was something else? Also, the authors simply provide a color for each bait protein following a color code pretended to reflect abundance. This is insufficient. At a minimum, they should provide at a supplementary table with the peptides identified and the total spectral counts for each protein.

Response: Thank you for pointing this out. We rewrote the figure legend to read:

Relative abundance of Bsd interacting proteins recovered from 12-24hr embryo lysates, determined by MS. See Table S2 for detailed quantification.

We have included detailed MS results in Table S2 per the reviewer's suggestion.

Comment: Fig S3A makes little sense. If the control is GST alone, it should be indicated. What is shown in lanes 2 and 3? In both, we see a strong band at 25 kDa, the MW of GST. Where is GST-Bsd? I would expect a band at a much higher MW. And if lanes 4 and 5 are like 2 and 3 but with the addition of embryo lysate, then why do we not see the 25 kDa band like in lanes 2 and 3? Is the elution different between 4 and 5 vs 2 and 3?

Response: We agree. The original blot was mislabeled!! Thank you for finding this error. We have labeled the blot correctly in the revised manuscript, and highlighted the lanes used for MS analysis.

Comment: The data presented in Fig 3B is inconclusive. Bsd-Myc and Bsd.Q545*-Myc should be shown on the same membrane to compare levels (both from the extracts and the IP products). A Western of the Polo-Flag IPed should also be shown. The same requirements apply to results shown in Fig 3C and 6E.

Response: It seems the blot images were over-cropped in the original submission. We replaced the images in question so that more of the membrane can be seen (with both inputs now clearly visible on the same membrane). Thank you for the suggestion.

Comment: Fig 4A and C: The total IPed protein (Polo-Flag or Polo-GFP) should be blotted to compare phosphorylated / total.

Response: We tried to visualize total protein IP'd using the affinity tags, but this approach did not work in our hands. We added a Western Blot of total protein lysate using anti-pThr210-PLK1 (Fig. S4C), which supports the results in Fig. 4A,C by showing Bsd transfected cells had more Polo^{pT182} than control cells. In general, only inputs are shown for IP experiments, even for more complex 'competition IPs' (e.g. PMID:25119050 Fig3A).

Comment: Fig 4B: PoloT182A + Bsd should be tested in parallel and results shown.

Response: Great suggestion!! PoloT182A failed to translocate to the nucleus in cells transfected with Bsd (revised Fig. 4B)

Comment: Fig 4D: Which antibody was used to stain pPolo? This is nowhere indicated. I hope it's not the generic pThr antibody.

Response: The pThr210-PLK1 antibody was used to visualize Polo^{pT182} in vivo. Since this was unclear, we changed the Fig. 4D label from 'pPolo' to 'Polo^{pT182}'. Thank you for pointing this out.

Comment: Movie 2 needs a WT control for comparison.

Response: Done!

Comment: Figure S4B does not show an in vitro phosphorylation assay as indicated in the legend. It is simply a Western from an IP.

Response: We have rewritten the legend. Thanks!!!

Comment: The last part of the manuscript attempts to connect Polo with Tumbleweed and Pavarotti as effectors in the regulation of the cytoskeleton during myotube formation. Unfortunately, the evidence presented is far from sufficient in many ways. First, there is no previous evidence that Tumbleweed or Pavarotti is a phosphorylation target of Polo regulated in their function by Polo in *Drosophila*. This manuscript does not provide any such evidence either.

Response: The reviewer makes a good point. Polo directly binds Tum, and Polo regulates Tum/Pav localization (PMID: 20516152, 20628062). The previous studies did not show Polo phosphorylates Tum or Pav, and it is conceivable that Polo regulates Tum/Pav localization through a phosphorylation independent mechanism. We used transgenic Tum-GFP to assay Tum localization in myotubes prior to our original submission. Unfortunately, this tool was not of sufficient resolution to visualize Tum localization *in vivo*. Characterizing Tum phosphorylation by Polo in vitro would not necessarily answer whether Tum is a Polo effector

during myogenesis, so we took a genetic approach to understand if Polo and Tum act in a common myogenic pathway (see response below).

Comment: Suppositions are made based on what is known about mammalian orthologs in cytokinesis. But they would need to be tested here in myoblasts in *Drosophila*. The implication of Tumbleweed in myotube formation is limited to the data presented in Fig 4G, H. No biochemical or genetic link is established with Polo and nothing at all is showed about Pavarotti.

Response: To strengthen the hypothesis that Tum/Pav are Polo effectors during myogenesis, we performed double mutant analysis. Embryos homozygous for *tum^{DH15}* or *polo¹* showed myotube guidance defects, and *tum^{DH15}* embryos showed the stronger myogenic phenotype. The phenotype in *tum^{DH15} polo¹* double mutant embryos was not significantly different than *tum^{DH15}* embryos (Fig. S6A,B). Since the phenotype in *tum^{DH15} polo¹* embryos was not an additive effect of the single mutant phenotypes, we can conclude Polo and Tum act in a common pathway. We have also included *pav^{B200}* embryos in the revised manuscript, and the myogenic phenotype is consistent with a role for Pav in regulating myotube guidance (Fig. S6C). Since Polo-Tum and Tum-Pav physical interactions are direct and the binding domains have been mapped (yeast two hybrid), further biochemical studies are not likely to be informative without significantly increasing the scope of the study.

Comment: Also, I am not convinced that the foci seen with the γ -Tub antibody in Fig 5D and S5 are specific or meaningful in any way. They simply seem to be everywhere in the cytoplasm.

Response: The use of the γ -Tub antibody was first reported for *tum^{DH15}* myotubes in (PMID: 19297411). We improved upon the previous study (1) by marking individual LO1 myotubes with *slou>GFP* and (2) by quantifying individual foci. Our quantification clearly shows a transition in γ -Tub localization, which confirms the conclusions reported in (PMID: 19297411) and supports our results from the *Nod.GFP* experiments.

Comment: No statistics are done to analyzed the data presented in Figs 1B, F, G, 2B, 3F, G, 4F, H, 5C, 6C, S3B, C. It seems that experiments were done only once.

Response: We apologize for the confusion. Our muscle heat maps were intended to replace histograms quantifying phenotypic frequencies for 30 muscles in each genetic background. Each heat map is derived by assessing muscle morphology in 6 hemisegments per embryo and 9+ embryos per genotype. We added a section in the Methods to clearly explain our analysis and performed a Fisher's exact test or One-way Anova on all of the experiments in question. The statistical results are now shown in revised figures.

Comment: I don't understand the QQ plots shown in Fig S6C. What are the values being compared? What are the units on the axes?

Response: We again apologize for the confusion. Since we added a supplemental figure to address other comments, Fig S7 legend now reads:

The QQ plot of myotube length distribution, related to Figure 7C. Each data point represents the actual length distribution value (x) and the predicted length value for a normal distribution (y). The lengths of the con siRNA and DMSO treated myotubes fit the normal distribution; the lengths of the Vrk3 siRNA and Volasertib treated myotubes did not fit the normal distribution.

Comment: The model shown in Fig S6D is extremely premature.

Response: Thank you for pointing this out. Since we added a supplemental figure to address other comments, the model in Fig S7D was modified to show the interactions we identified are not necessarily direct.

Comment: Lines 190-192: It is written: “The Polo/Tum/Pav cytoskeletal regulatory module interacts with the microtubule cytoskeleton in post-mitotic myotubes, suggesting microtubules are the major target of this cytoskeletal regulatory module (Fig. S6D).” What interaction are the authors referring to? No such data is shown in the manuscript. Is this statement referring to a previous paper? In this case, there would be a reference missing?

Response: Thanks for catching this. We rewrote our conclusion to read:

Bsd is thus an essential regulator of the microtubule cytoskeleton (Line 242).

Comment: Line 25. I’m not sure what “mitotic growth” means. Mitosis and growth are separate processes. I suggest replacing with “mitosis”.

Response: Done! We have made the suggested edit throughout the revised manuscript.

Comment: Lines 95-96: The term “Affinity Purification and Mass Spectrometry (AP-MS)” is used incorrectly as it always refers to the purification of a tagged protein and the identification of the proteins co-purified. Here, the authors are doing GST pulldowns and they should call it that way.

Response: Done!

Comment: Lines 137-8: The increase in pHH3 cells upon expression of AurB is interpreted as an increase in mitotic index. However, as Ser10 of pHH3 is a direct target of AurB, the increased phosphorylation does not necessarily reflect a mitotic state. Conclusions should be adjusted accordingly.

Response: Thanks for this insight!!! We felt this was an essential point, so we used the FUCCI system to assay cell division in the embryonic mesoderm (Fig. 5E,F), and found AurB

increased the percentage of cells in G2/M.

Comment: Fig S4D: The distance units should be indicated.

Response: Done!

Comment: Lines 221-223: I don't understand the logic behind this statement: "In fact, the conservation of Bsd/Vrk3 cellular and molecular functions is so striking that Vrk3 likely regulates Plk1 activity under a variety of developmental and homeostatic contexts."

Response: Thank you for pointing out this issue. We rewrote the statement to read:

Our study highlights the exciting possibility that the role of Plk1 during vertebrate muscle development and regeneration extends beyond the mitotic events of myogenesis. (L353-355).

Reviewer #2

Comment: This manuscript describes an extensive piece of research that embraces an impressive array of experiments/assays and there is a lot of very good content woven into a satisfying narrative.

Response: We greatly appreciate the kudos. This project has been a lot work!! But it has also been a lot fun.

Comment: The novel suggestion presented here is that a mitotic kinase (Polo) with a documented function in the control of cell division also has a role in post-mitotic cells in aspects of cell differentiation (“cell guidance” in the authors’ turn of phrase). The question is then how does Polo affect cell divisions and subsequently cell differentiation? From the title onwards, the authors emphasise the spatial/temporal expression aspect of the expression of the Polo regulators Bsd and Aurora B in order to explain.

Response: Thank you for recognizing that our surprise result has been a challenge to understand, and a challenge to explain in light of the well-known mitotic roles for Polo. Your comments for improvements are greatly appreciated!

Comment: There are very different cytoskeleton behaviours in cell division and cell guidance. One might anticipate that possibly some similar players might be involved in the co-ordination of these different processes, although I am not aware of a previous specific implication, as made in this study, of a specific player like Polo. This study therefore has a core content of real interest (and not just for muscle biologists).

Response: Thank you for validating our work. We had hoped this work would be well received by a broad audience. Fingers crossed!

Comment: In general I feel that more detail and clarity on aspects of the methodology and the presentation of the results obtained would be helpful for the reader and would further improve the m/s. A key point is to be more precise about what is happening when and where in terms of expression of Bsd, Polo and AuroraB, in relation to cell division and cell morphogenesis.

Response: We appreciate the insights and hope our revisions have broadly addressed these concerns.

Comment: Bsd expression. One important point is how clear is the expression data to support the statement in the Introduction (I57): “the transition from mitotic growth to cellular morphogenesis is achieved through the spatially and temporally restricted expression of the Aurora kinases and Bsd”.

Response: Thank you for taking a close look at the expression data. With respect to AurA/B,

we did fluorescent in situ hybridization (we requested the antibodies, but the pandemic has caused some issues with delivery it seems). We detected high levels of each mRNA in the somatic mesoderm at Stage 10, which correlates with H3 phosphorylation in the somatic mesoderm (we made a new figure for all of the Aurora-related revisions, Fig. 5A shows the in situ). By Stage 12, *aurA* and *aurB* mRNA expression is restricted to the CNS as is phosphorylated H3.

Comment: Start with Bsd. L79 “Bsd is ubiquitously expressed in blastoderm embryos, but after gastrulation Bsd expression in the mitotic mesoderm was significantly reduced”. It could be, but no measurements are presented, and the embryo images are small and mesoderm is not labelled making assessment by the reader difficult, especially for the non-specialist.

Response: We modified the Bsd expression figures for clarity, and rewrote our results for the Bsd expression pattern. The full text now reads:

We generated and validated an antibody against Bsd (Fig. S2A-C), and found Bsd was ubiquitously expressed in blastoderm embryos (Fig. S2D). However, after gastrulation, Bsd expression became dynamic. Founder cells are specified during Stages 10-11, and Bsd expression in the mesoderm during Stage 11 was reduced compared to subsequent stages of myogenesis (Figs. 2A, S2D-F). Myotube guidance begins at Stage 12, and Bsd expression levels in the mesoderm peaked during Stages 12-14 (Figs. 2A, S2D-F). Myotube guidance is complete by Stage 16, and relative Bsd expression levels in the mesoderm were reduced Stage 16 embryos (Figs. 2A, S2D-F). The Bsd expression pattern is consistent with our hypothesis that kinase expression is spatially and temporally regulated during embryogenesis. (Lines 112-120).

Comment: L81 “Bsd expression became progressively enriched”. It is difficult for the reader to see this from what is presented. The plot in a supplemental figure (Fig S2 E) is the clearest to show an increase in nuclear Bsd between stages 11 and 14, but even this is dependent on a constant level in the non-mesoderm nuclei (is this known to be the case?). The authors should explain the evidence that lies behind using this ratio of intensities.

Response: This is a good point. The comparison is more meaningful within the mesoderm, so we quantified relative expression between Stages 11, 12, 14, and 16 in the *Mef2+* cells of the mesoderm. The new graph is now shown in Fig. S2E.

Comment: The main Fig 2A is not compelling as support. Stage 14 expression looks less than stage 12, but Fig 2SE says it is higher.

Response: Agreed. We added a heat map images of Bsd fluorescence, which clearly shows highest expression at Stage 12. Our new graph (Fig. S2E) agrees with this observation. Thank you.

Comment: The Fig 2 legend also refers to nuclear/cytoplasmic distribution. This is not clearly

shown, not least taking into account the different distribution of nuclei in the myogenic mesoderm at stage 11, compared to stages 12 and 14.

Response: Fair point. We removed the sentence. In future studies we hope to show functional data for nuclear and cytoplasmic functions of Bsd.

Comment: Bsd muscle phenotype. In Fig 1 the authors have taken a thorough approach to the challenge of analysing and presenting a relatively mild (and variably penetrant) muscle phenotype. However, the muscle phenotype scoring, especially for readers not familiar with the *Drosophila* embryo musculature, could be more completely explained. Maybe because of this, I had difficulty in relating the heat map of Fig1B to the plot in Fig1G.

Response: It seems we were overly brief in our initial submission. We have added a section to the Methods explaining our phenotyping in more detail (***Phenotypic scoring, analysis, and visualization***). We also added a paragraph to the Results to clarify how the muscle pattern in Fig. 1B develops (Lines 85-96, and we have clarified the heat maps and histologic scoring plots in the figure legend. Our phenotype may seem mild, but we recently published a follow-up screen in which we hit *bsd* two more times (PMID: 33993253).

Comment: Fig 1 legend says “individual muscles were scored in segments A2-A8 of st 16 embryos”. Each abdominal segment A2-A8 contains the same stereotypic muscle pattern and there are two of each muscle (e.g. muscle VL1) in each segment, one on each side of the midline, i.e. 14 examples of each muscle in A2-A8 per embryo. However, if the authors actually mean the more usual hemi-segments, then this number reduces to 7 per embryo.

Response: Agreed! We have used the term hemisegment throughout the revised manuscript and double-checked that our text accurately reflects our quantifications.

Comment: The plot in Fig 1G says it shows the sum of phenotypic scores divided by number of embryos. To readily interpret this, it should say how many muscles were scored per embryo. It would be 7 if 7 hemi-segments were scored, 14 if 7 segments. Thus, a “histologic score” of 7, e.g. for muscle VT1, would mean an average score of 1 per muscle (if 7 muscles per embryo).

The score of “3” for VL1 would mean an average score of 0.43 per muscle. In the scoring system used, muscle VL1 would thus not be missing, very rarely would have its elongation affected, and a little less than half of them would have attachment affected. However, the shade of blue on the heat map in Fig 1B suggests VT1 is only affected in 20-25% of examples, so maybe the number of muscles per embryo the authors are using is 14, i.e. they count segments.

Response: Thank you for pointing out that the explanation of our Histologic Score was unclear. The Histologic Score was applied to data using *5053>GFP* and *slou>GFP* to mark specific muscles. Comparisons can't be made between Fig 1G and Fig 1B because the data in Fig 1B used Tropomyosin to visualize all muscles. We have made this point clear in the

revised figure legend. The discrepancy between Fig. 1B and Fig 1G highlights that our *5053>GFP/slou>GFP* labeling is more precise in assessing phenotypes.

We have also included the calculation for the Histologic Score in our revised Methods, which now reads: 'Histologic Score' was calculated using the following scale: missing=3, elongation defect=2, attachment site defect=1, normal=0. Histologic Score = sum of phenotypic score/number of embryos analyzed as described [67].

To directly answer your questions for the VT1 muscle in *bsd* embryos, our calculation is as follows:

Histologic Score: normal=30*0=0, attachment site defect=13*1=13, elongation defect =14*2=28, missing=15*3=45, 15 embryos scored

Histologic Score=(0+13+28+45)/10=8.6 (see Fig. 1G)

Comment: Which step of muscle differentiation does *bsd* affect? In order to conclude that *bsd* affects muscle cell guidance, an important point is that it doesn't affect a step in the muscle development pathway prior to this. The authors addressed this by asking about muscle precursors/founders. However, only a supplemental figure (Fig S1E,F) is presented to support the statement (I71) that "Bsd is not required for muscle precursor specification". The images are not the highest quality, but they are quantified and this shows that the number of muscle precursors is unchanged. It would help the reader, especially the non-specialist, if they used the same terminology. Thus, they refer to muscle "precursors" in the text (I72), but to "founder cells" in the figure legend.

Response: We largely agree with the reviewer and we dedicate extensive time toward distinguishing cell fate specification, fusion, and guidance phenotypes for each mutant we characterize. One important point is that normal guidance can occur in a fusion mutant, but the mutant muscle will be thin (we discuss this in PMID: 33993253). There are also instances where both guidance and fusion are affected in a particular mutant background, which makes sense since both processes involve regulated changes to the cytoskeleton. In response to this comment, we have used improved images in Fig. S1E and consistently use 'founder cells' throughout the text.

Comment: It would also help if the use of *5053>GFP* and *slou>GFP* was explained. These markers together only address 4 of the 30 founder cells per hemi-segment. Moreover, they don't mark any of the founders for the five muscles that are most affected in the *bsd* mutant in Fig 1B (LT4, DO3, DO5, VA3, DA3). A marker for all founders (e.g. *rp298*) could be used.

Response: These are great suggestions, and we have expanded our explanation of *5053>GFP* and *slou>GFP* in the Results (Lines 97-105). We did the *rp298>GFP* experiments early in the project. Unfortunately by the time *rp298>GFP* is robustly expressed, the founders have developed into myotubes and quantifying 30 myotubes in 7 hemisegments turned out to be crude and imprecise. We developed the *5053>GFP/slou>GFP* approach so that we could accurately detect and follow a subset of founders throughout myogenesis. What we have

shown is that for some muscles (e.g. LO1), the founder is correctly specified in *bsd* embryos and that myotube guidance is subsequently affected (Movie 1). From this model we can predict that similar events occur in the remaining muscles. We screened the Janelia collection for ‘identity gene’ Gal4 lines to visualize other muscles, but only *slou.Gal4* reliably marked both founder cells and myotubes. Nonetheless the *5053>GFP/slou>GFP* approach allowed us to visualize myotube guidance at single cell resolution.

Comment: Polo. It would be helpful to include more information on the polo hypomorph allele. Does it have a phenotype prior to muscle cell guidance? One would expect many polo mutants to have earlier phenotypes that might influence how muscle develops, e.g. affect an aspect of cell division that could affect the myoblast pool available for muscle development. As for *bsd*, the question also arises as to whether polo affects muscle precursor/founder specification. A gap in the manuscript is that only a stage 16 (near the end of embryogenesis) muscle phenotype is shown.

Response: Thank you for the suggestions. We had been operating under the assumption that maternally contributed *polo* allowed *polo*¹ embryos to develop normally until myotube guidance initiates. In response to this comment, we assayed founder cell specification in *polo*¹ embryos and found *5053>GFP* and *slou>GFP* expressing founders were correctly specified (Fig. S3E,F). To clarify how the *polo*¹ allele affects early embryonic development, we have added the following text:

The *polo*¹ allele is a maternal effect mutation that disrupts mitotic chromosome alignment [46], but zygotic mutants are partially viable and do not show mitotic defects until after embryogenesis [47]. (Lines 148-150).

Comment: Mesoderm mitotic cell divisions. When considering the role of Polo plus AurB in cell division, and Polo plus Bsd in muscle cell guidance, important aspects are when are mitotic cell divisions complete in the myogenic mesoderm, and what is the expression of these three players with respect to this time. The Model in Fig 6G indicates mitosis is complete in the mesoderm by before stage 11. However, although this is the case for the third mesodermal cell division, the fourth mesodermal division continues in stage 11 and 12. (Bate (1993), The mesoderm and its derivative, pp1013-1090, in The Development of Drosophila melanogaster, CSHL Press). Indeed, FigS4E shows myogenic mesoderm cell division still continues in stage 11.

Response: Fair points. As explained previously, we now show AurA/B mRNA expression in the mesoderm in parallel to pH3. The reviewer and Michael Bate are of course correct that mesoderm cell division continues into St12, but the number of Mef2+ pH3+ cells is dramatically reduced in St12 embryos compared to St10 embryos (Fig. 5C).

Comment: L135 “a majority of cells in the mesoderm exit the cell cycle prior to the onset of myotube guidance”. This implies that the transition between cell division and morphogenesis in the whole population is not uniform, and there is need to assay on a cell-by-cell basis in

order to establish the expression of these players with respect to the change from mitosis to muscle cell differentiation.

Response: We agree with the reviewer that our language was imprecise. In response to this comment, we double labeled *rp298.nlacZ* embryos for lacZ and pH3 and did not see any double positive cells in St12 embryos (Fig. S5A). This result argues all nascent myotubes are post-mitotic when myotube guidance initiates. We have modified our text to read:

After gastrulation, the mesoderm undergoes four rounds of cell division that largely conclude by the onset of myotube guidance [49]. Nascent myotubes exit the cell cycle and begin myotube guidance during Stage 12 (Fig. S5A), so it was surprising that the mitotic kinase Polo would be activated in the Stage 12 mesoderm (Fig. S4E). (Lines 185-188)

Comment: Aurora expression. L115 – “since AurB and AurA are not expressed in post-mitotic mesoderm during muscle morphogenesis”. This is important for the authors’ hypothesis for how the control of Polo kinase activation changes from AurB during mesodermal mitosis to Bsd during muscle cell guidance. The support for this idea needs more precision. When is the last mitosis in the myogenic mesoderm? (see above) What is the evidence that there is no mesodermal Aurora expression after this point? The authors just cite refs 44 and 45 in support. Both are in situ hybridisations (not protein) with little detail. 45 is the bulk assay of gene expression in the Drosophila genome project. These citations do not seem sufficient for this important point. This really needs the authors to include their own high quality study specifically addressing the question of the timing of any Aur B (and A) expression in the mesoderm.

Response: Great suggestion. As stated previously, we did fluorescent in situ hybridization for *aurA* and *aurB*. We detected high levels of each mRNA in the somatic mesoderm at Stage 10, which correlates with H3 phosphorylation in the somatic mesoderm (we made a new figure for all of the Aurora-related revisions, Fig. 5A shows the in situ). By Stage 12, *aurA* and *aurB* mRNA expression is restricted to the CNS as is phosphorylated H3. We have been more explicit about the timing of the cell divisions in the Results, and clarified cell cycle exit has occurred in nascent myotubes by Stage 12.

Comment: In the Introduction (I52) it would help the reader to know here when mitosis finishes in the developing mesoderm, before (s)he considers when different players in this study are expressed.

Response: No problem. We have rewritten the Introduction to read:

Here we report a novel function for Polo in the post-mitotic mesoderm, where Polo regulates cellular morphogenesis. The embryonic mesoderm undergoes multiple rounds of cell division after gastrulation, and mitosis in the mesoderm is largely complete by Stage 12. Nascent myotubes exit the cell cycle and begin myotube guidance during Stage 12, and we found Polo is activated in the post-mitotic mesoderm of Stage 12 embryos. Aurora kinases are the

known activators of Polo, but the Aurora kinases were expressed in the embryonic mesoderm only until Stage 10. In contrast, the expression of the kinase Back seat driver (Bsd) was enriched in the Stage 12 mesoderm, where Bsd promoted Polo activation and directed microtubule reorganization necessary for myotube guidance. These studies argue the transition from mitosis to cellular morphogenesis is achieved through the spatially and temporally restricted expression of the Aurora kinases and Bsd. (Lines 57-67).

Comment: I73 states “Bsd directs myotube elongation and muscle attachment site selection”. This conclusion would be strengthened by using a marker for attachment sites. The question is: have the attachment sites changed position, or do the developing muscles miss their usual site and attach elsewhere? If the attachment site positions change, this questions the mesoderm cell autonomous function of *bsd*.

Response: Agreed. We had characterized tendon cells in *bsd* mutants using *sr>GFP* prior to the initial submission, but didn't see anything significant. Since *Mef2.Gal4* rescued the *bsd* phenotype, we felt the *sr>GFP* data wasn't necessary to establish a cell autonomous function. But moving attachment sites was a particularly interesting idea. We did live imaging of tendon cells but don't see any obvious translocations (Movie 2).

Comment: I92 (and Fig 2): is the Bsd.I129A mutant expressed (at the same level as the WT)? This needs to be shown in order to conclude it doesn't have the effect of the WT.

Response: Fair point. We had used the attB vectors to target wild-type and I129A to a common integration site at 22A2 using ϕ C31 (see Methods). In response to this comment, we did qPCR on each line and found the difference in mRNA expression was not significantly different between wild-type and I129A insertions (Fig. 2D).

Comment: I100 says “the truncated Bsd.Q545* protein”. This would be useful to include in paragraph 1 of the results.

Response: Done!

Comment: Fig 1D: wt and *bsd1* are labelled the wrong way around.

Response: Fixed!

Comment: Fig 1F: the histogram is not labelled to indicate which muscle the results are from.

Response: Fixed!

Comment: From the Mass spec results (Fig 3A), explain why Polo was pursued (it is low down on the presented list).

Response: No problem. We had screened candidates for protein-protein interaction with Bsd

in S2 cells and checked candidate alleles for muscle phenotypes. Polo was the only candidate with a validated interaction and a phenotype. Here is the revised text:

To uncover Bsd effectors during myotube guidance, we used recombinant Bsd to precipitate Bsd interacting proteins from whole embryo lysates, which were then sequenced by Mass Spectrometry (MS; Figs. 3A, S3A). The MS results identified over 150 candidates that interacted with Bsd, none of which were known to regulate muscle morphogenesis (Table S2). Since the absolute quantity of each candidate in the input embryo lysate was not known, the relative abundance of a candidate in the pool of precipitated proteins could not be used to prioritize the potential Bsd-interacting proteins. To rank the candidates for further analysis, we first validated protein-protein interactions in S2 cells, and then characterized muscle morphogenesis in embryos with reduced candidate gene function. The mitotic kinase Polo showed the strongest interaction with Bsd in S2 cells (Fig. S3B), and we confirmed the interaction with reciprocal immunoprecipitation experiments (Fig. 3B,C). Strikingly, the Bsd.Q545* protein encoded by our EMS-induced *bsd¹* allele did not interact with Bsd (Figs. 3B,C). Embryos homozygous for the hypomorphic alleles *polo¹* and *polo^{KG03033}* showed myogenic phenotypes similar to *bsd* embryos (Fig. S3C,D). We tested four candidates in addition to Polo, and although these proteins showed a weak interaction with Bsd in S2 cells (Fig. S3B), we could not confirm muscle phenotypes with alleles that affected the remaining candidates. Polo was thus the highest-priority candidate for further analysis. (Lines 130-146)

Comment: l122: what stage are the embryos in the experiment that shows less in vivo Polo phosphorylation in *bsd* mutants.

Response: Stage 12. Revised.

Comment: Fig5: explain what LifeAct.RFP is.

Response: Done!

Comment: References: A sentence in the Introduction (l48) as written seems to imply that the references 28-31 support the comment that “muscles in *Drosophila* are easily visualised in live, unperturbed embryos”. These references are not for live embryos.

Response: Thanks. We modified the references for precision.

Comment: Methods: The authors say in three sections “include” when listing reagents used (l233, l267, l278). This gives the impression that not all reagents are listed.

Response: The text was changed from ‘include’ to ‘were’.

Comment: Conclusion/Interpretation. Overall, it didn't feel that the experiments undertaken and the results obtained were always matched by the thoroughness of the explanations and descriptions. This should be readily addressable. The manuscript needs to be tightened up in

some of the conclusions the authors emphasise (see above).

Response: Once again, it seems we were overly brief. We have expanded our explanations in multiple places and hope we have addressed your concerns in general.

Comment: Clarity and context. Title: Do the expression results presented support the title? Abstract (I11) says “The transition between mitotic growth and morphogenesis is accomplished through the spatiotemporal transcriptional regulation of AurB and Bsd”. However, “transcriptional regulation” has not been shown. In the introduction (I59) the authors just say “restricted expression”. Even described like this, how clearly has this been shown? (see other comments).

Response: Thanks for highlighting these issues. We have revised our word choice throughout the manuscript. We certainly concede that transcriptional regulation was not shown. But in the revised manuscript we now show the detailed expression patterns of 3 regulatory kinases, and feel we have now supported our title.

Comment: The first paragraph of the Discussion is a succinct, clear, accurate summary. Some of the other sections could follow a similar presentation.

Response: Thanks for the honest appraisal. We have made extensive edits to the text and hope we have delivered our message clearly in the revised manuscript.

Reviewer #3

Comment: The paper is clearly written with beautifully presented data. Overall, the work is novel as there are few post-mitotic roles for Polo and the paper is of interest in the field of muscle development.

Response: Thank you for your interest in our work, and for the kind assessment of our data!!!

Comment: Table S1 is missing.

Response: Sorry for the confusion. Tables S1 and S2 are Excel documents and should be available as additional files. Fingers crossed they are now accessible!!! If not, please ask for these essential documents to be made available.

Comment: There does not appear to be much difference in Bsd mesoderm staining from stages 13 to 16 to support the claim that 'but during the final stages of myogenesis Bsd expression in the mesoderm was again reduced.' (p. 4, lines 82-83). Even in Figure 2A the expression of Bsd seems present in stage 16, just in the cytoplasm (and possibly slightly enriched in nuclei at muscle ends).

Response: The reviewer is correct that the change from St14-16 is modest. We have reassessed the expression data and present new panels (heat map in Fig. 2A and revised quantification in S2E) that better depict the changes in expression.

Comment: Are *bsd* mutants embryonic lethal?

Response: Yes!!! We added a sentence clarifying that issue in the Results (Line 76).

Comment: Does loss of *bsd* also affect myoblast fusion? Panels 1E, 2B, 4F, 4G and possibly panels S1B (small pictures) show unfused myoblasts. siRNA of *Vrk3* (panel 6A) also shows fusion defects. If true, these data suggest that *Bsd* and *Vrk3* are required for an additional post-mitotic function. This should be clarified.

Response: Impressive analysis of our micrographs. The small rounded cells could be unfused myoblasts, or they could be multinucleate 'myobags' that failed to elongate. We now report the fusion index after the first round of fusion in Stage 12 embryos (Fig. 1H). The *bsd* fusion index is 85% of controls, so there is a modest but statistically significant fusion defect.

Comment: Does *ballchen* have embryonic muscle morphogenic defects? There are reported muscle defects for *ballchen* RNAi knockdown later in development, so could it be partially redundant with CG8878? Note that many muscle looks OK in *bsd* mutants.

Response: Great point. There may be some redundancy between the paralogs. However, the null allele *ball*² (PMID: 24876388) and the alleles *ball*^{E24}, *ball*^{E107}, *ball*^{E60}, and *ball*^{trip} (PMID:

16301329) are reported to be larval or pupal lethal, arguing embryonic muscle development was normal enough for the mutants to emerge from the chorion. The embryonic lethal alleles *ball*⁴³ and *ball*⁵³ are longer being maintained (we requested them!!!), but based on the more recent *ball*² results, *ball*⁴³ and *ball*⁵³ embryonic lethality is likely due to second site mutations. We added the following sentence to the Results:

The Bsd paralog Ballchen (Ball) regulates sarcomere assembly in adult flight muscles, but the null allele *ball*² is embryonic viable [38], suggesting Ball is not a major regulator of embryonic muscle development. (Lines 80-82)

Comment: In the AP-MS experiments, were other candidate protein interactions verified? Via the heat map, Polo was not as strong as some other candidates.

Response: Agreed, Polo was pretty far down the list. However, the heat map does not necessarily correlate with the strength of the interaction. For example, a protein with low expression may have relatively few reads due to the reduced availability of the protein for an interaction. But in response to this comment we have included our rationale for following up on Polo rather than the other proteins and made a new table (Table S2) detailing the MS results. Our revised texts reads:

To uncover Bsd effectors during myotube guidance, we used recombinant Bsd to precipitate Bsd interacting proteins from whole embryo lysates, which were then sequenced by Mass Spectrometry (MS; Figs. 3A, S3A). The MS results identified over 150 candidates that interacted with Bsd, none of which were known to regulate muscle morphogenesis (Table S2). Since the absolute quantity of each candidate in the input embryo lysate was not known, the relative abundance of a candidate in the pool of precipitated proteins could not be used to prioritize the potential Bsd-interacting proteins. To rank the candidates for further analysis, we first validated protein-protein interactions in S2 cells, and then characterized muscle morphogenesis in embryos with reduced candidate gene function. The mitotic kinase Polo showed the strongest interaction with Bsd in S2 cells (Fig. S3B), and we confirmed the interaction with reciprocal immunoprecipitation experiments (Fig. 3B,C). Strikingly, the Bsd.Q545* protein encoded by our EMS-induced *bsd*¹ allele did not interact with Bsd (Figs. 3B,C). Embryos homozygous for the hypomorphic alleles *polo*¹ and *polo*^{KG03033} showed myogenic phenotypes similar to *bsd* embryos (Fig. S3C,D). We tested four candidates in addition to Polo, and although these proteins showed a weak interaction with Bsd in S2 cells (Fig. S3B), we could not confirm muscle phenotypes with alleles that affected the remaining candidates. Polo was thus the highest-priority candidate for further analysis. (Lines 130-146)

Comment: Is it conclusive that AurA and AurB are not expressed in mesoderm tissue? Even if RNA in situ data shows decreased levels in mesoderm (per Flybase), there may still be some faint signal. The Flybase Developmental proteome data also shows relatively high protein expression for AurA and AurB after stage 12 (i.e., is this nervous system specific or is there some protein expression in muscle tissue)? This needs to be established as a major tenant of this work is the idea that AurA switches it Polo activation role to Bsd.

Response: Great suggestion. We did fluorescent in situ hybridization for *aurA* and *aurB* and double-labeled with Mef2 (we requested Aurora antibodies, but the pandemic has caused some issues with delivery it seems). We detected high levels of each mRNA in the somatic mesoderm at Stage 10, which correlates with H3 phosphorylation in the somatic mesoderm (we made a new figure for all of the Aurora-related revisions, Fig. 5A shows the in situs). By Stage 12, *aurA* and *aurB* mRNA expression is restricted to the CNS as is phosphorylated H3.

Comment: What are explanations for the lack of complete rescue of *bsd* mutants by expression of PoloT182D? Does expression of PoloT182D on its own give a phenotype?

Response: *Mef2>T182* embryos do show a phenotype. We have changed Fig. 4 legend to read:

Note that expressing in Polo^{T182D} otherwise wild-type embryos caused a modest muscle phenotype, which may explain the incomplete rescue of the *bsd'* phenotype.

We have added a paragraph to the Discussion highlighting other potential targets of Bsd that reads:

We found that activated Polo partially rescued the *bsd'* embryonic phenotype (Fig. 4F). While the levels of activated Polo in the rescue experiment may not have recapitulated endogenous levels of activated Polo, it is also possible that Bsd regulates Polo-independent pathways. For example, we validated a physical interaction between Bsd and the Myosin VI protein Jaguar (Jar; Fig. S3B). Jar stabilizes interactions between the actin and microtubule cytoskeletons [67], and is required for cell migration [68] and cellular guidance [69]. Although one essential role for Bsd is to activate the Polo/Tum/Pav cytoskeletal regulatory module, it seems likely that Bsd regulates additional cytoskeletal proteins, such as Jar, to direct myotube guidance. (Lines 313-321)

Thanks for the suggestion!!!

Comment: The experiment showing that misexpression of AurA by *mef2-Gal4* prolongs mitosis is intriguing, but does not prove that activation of Polo switches from AurA to Bsd. What if AurA is indeed expressed, but held inactive and/or regulated, in muscle tissue under normal conditions. Overexpression of AurA may simply overcome this inhibition. What if activated Polo is expressed in AurA and/or AurB backgrounds?

Response: Thank you for pointing this out. We agree that additional functional data was needed. We checked activated Polo (Polo^{P^{Thr182}}) in the mesoderm of Stage 12 *aurA* and *aurB* mutants, but there was difference between mutants and controls (Fig. 5D). Also, there was no muscle phenotype in *aurA* or *aurB* embryos (Fig. S5B). These data suggest that AurA and AurB are not required to activate Polo during muscle morphogenesis.

Comment: Both Tum and Pav have been shown to be Polo effectors. Only Tum is examined in this manuscript. Thus, some sentences need to be modified to focus only on Tum (p., 7, lines 149-150; p. 8, line 163; 9, line 190).

Response: Fair point. We have included the *pav* mutant phenotype in the revised manuscript (Fig. S6C). *pav* muscle morphogenesis defects are similar to those we identified for *bsd*.

Comment: How exactly is Polo directing MT organization? Multiple reports show that Polo directs MT nucleation. This does not appear to be discussed or specifically tested (other than showing that gamma-tubulin is aberrant, which could be direct or indirect).

Response: Great suggestion!! We included a paragraph in the Discussion outlining the known roles of Polo in microtubule nucleation during mitosis and how we envision Polo regulating microtubules during myogenesis (Lines 322-342). It seems the answers to this question will require structure/function rescue experiments with proteins lacking NLS or NES, which could be fun for future work.

Comment: Can Vrk3 substitute for Bsd to rescue muscle morphogenesis defects?

Response: Great question. We have recovered Vrk3 transgenic insertions that we are crossing into the *bsd*¹ mutant background. Fingers crossed for a rescue.

Comment: The statement on p. 8, line 162 is not quite accurate. Bsd is not an essential regulator of MY dynamics. Features of dynamics (treadmilling, etc) were not tested.

Response: Agreed. We have removed *dynamics* from the revised manuscript and use the term *MT reorganization* to describe the function of Bsd throughout the manuscript.

Comment: Guidance implies that factors influence direction and attachment site specificity. How does the Bsd↔Polo pathway fit into this paradigm? This must be discussed.

Response: We have added a paragraph to the Discussion that places Bsd into our overall model (Lines 297-312). Thanks!!!

Comment: The phrase 'in turn to the skeleton' doesn't read well. There also is needs to be a bit more in the introduction about the conservation between *Drosophila* and mammalian myotube guidance (p. 3, line 48). What is known? What remains to be uncovered?

Response: Got it. Rewritten.

Comment: Figure S1A – I believe 'Not data' should be 'No data available' underneath the graphs

Response: Thanks for catching that. Changed to No Data.

Reviewers' Comments:

Reviewer #1:

Remarks to the Author:

I have examined the revised manuscript by Yang et al. Although the study is interesting and definitely worth pursuing, it is still too inconclusive to deserve publication in my opinion. Although the authors have satisfactorily addressed some of my concerns, at least two of my essential major concerns remain:

1-The authors have not shown that Bsd directly phosphorylates Polo at T182. I understand that they are being careful with words to avoid drawing this conclusion explicitly, but this is still what they are leading readers to believe. The difficulty in expressing and purifying recombinant Polo from bacteria that they mention as an excuse is not insurmountable. For example, Carmena et al (2012, PLoS Biology, PMID 22291575) have expressed and purified HIS-Polo WT and T182A to show that Aurora B directly phosphorylates Polo in vitro (see Figure 4A in that paper). Until this crucial point is resolved, it will remain quite possible that Polo is still phosphorylated by Aurora B or Aurora A (or another kinase for that matter) in the context of muscle development studied here. The kinase activating Polo could simply depend on the activity of Bsd, either directly or indirectly.

2-The molecular mechanisms controlled by Polo in muscle morphogenesis need to be much better characterized. The direct substrate(s) of Polo in this process should be identified, and how its activity is modified by this regulation should be understood at a minimal level.

These elements of molecular mechanisms directly upstream and downstream of Polo in muscle morphogenesis would need to be established for the paper to put forward reliable conclusions and constitute significant advance, in my opinion.

Reviewer #3:

Remarks to the Author:

Reviewer #3's comments on responses to previous concerns

The manuscript by Yang, et al., is largely improved and addresses major reviewer comments. Minor comments that should be corrected are below:

- 1) p. 4: The correct elongation and attachment of individual muscle results in a stereotypical morphology for each muscle that forms a largely invariant pattern in abdominal segments A2-A8 (Fig. 1B,C) should refer to Fig. 1A and 1B (not C).
- 2) Fig. 2A: no preimmune blot is shown as indicated in the figure legend.
- 3) MW markers should be added to Western blots. Especially Fig. S2B, which states 'The Bsd and Myc antibodies recognized proteins of the same molecular weight.' While the myc tag is not large, you may be able to see a slight shift. Or rephrase this sentence.
- 4) Why is 24B-Gal4 used for rescue while qPCR results for *mef2-Gal4* are used (Fig 2)? These are not comparing the same things.
- 5) Please be consistent with I129 vs Iso129.
- 6) p. 6: The manuscript states that 'Bsd did not promote the phosphorylation of Polo serine residues,' but this is not shown.
- 7) Did *pav* enhance *tum* or *polo* mutant phenotypes? The experiment was only performed with *polo*, *tum* mutants.
- 8) Please add embryonic stages to the embryo panels that are present in supplemental figures as you did for the main figures. Easier to see than in the figure legends as there are many different stages.
- 9) There is no source information for *nod-GFP* or the anti- γ tubulin Ab. Please add.

Reviewer #3's further comments on responses to Reviewer #2's previous concerns

- 1) Comment: One important point is how clear is the expression data to support the statement in the

Introduction (157): "the transition from mitotic growth to cellular morphogenesis is achieved through the spatially and temporally restricted expression of the Aurora kinases and Bsd".

Response: By Stage 12, aurA and aurB mRNA expression is restricted to the CNS as is phosphorylated H3.

R#3 further comment: It is not obvious from the pictures presented in Figure 5A that AurA and AurB expression becomes restricted to the CNS. This reviewer cannot really see much expression in the CNS and hence this comment should be amended.

2) Comment: Polo. It would be helpful to include more information on the polo hypomorph allele. Does it have a phenotype prior to muscle cell guidance? One would expect many polo mutants to have earlier phenotypes that might influence how muscle develops, e.g. affect an aspect of cell division that could affect the myoblast pool available for muscle development. As for bsd, the question also arises as to whether polo affects muscle precursor/founder specification. A gap in the manuscript is that only a stage 16 (near the end of embryogenesis) muscle phenotype is shown.

Response: Thank you for the suggestions. We had been operating under the assumption that maternally contributed polo allowed polo1 embryos to develop normally until myotube guidance initiates. In response to this comment, we assayed founder cell specification in polo1 embryos and found 5053>GFP and slou>GFP expressing founders were correctly specified (Fig. S3E,F). To clarify how the polo1 allele affects early embryonic development, we have added the following text: The polo1 allele is a maternal effect mutation that disrupts mitotic chromosome alignment [46], but zygotic mutants are partially viable and do not show mitotic defects until after embryogenesis [47]. (Lines 148-150).

R#3 further comment: This clarification does not address the point of R#2. If maternal load precludes analysis of phenotypes during most of embryogenesis, how can the authors conclude that founder cell specification occurs normally (which happens in mid embryogenesis). Can you make maternal clones with a weak allele of polo? Can you demonstrate the indeed maternal load is present during stages 12-16?

3) Comment: What is the evidence that there is no mesodermal Aurora expression after this point? The authors just cite refs 44 and 45 in support. Both are in situ hybridisations (not protein) with little detail. 45 is the bulk assay of gene expression in the Drosophila genome project. These citations do not seem sufficient for this important point. This really needs the authors to include their own high quality study specifically addressing the question of the timing of any Aur B (and A) sexpression in the mesoderm.

Response: Great suggestion. As stated previously, we did fluorescent in situ hybridization for aurA and aurB. We detected high levels of each mRNA in the somatic mesoderm at Stage 10, which correlates with H3 phosphorylation in the somatic mesoderm (we made a new figure for all of the Aurora-related revisions, Fig. 5A shows the in situ). By Stage 12, aurA and aurB mRNA expression is restricted to the CNS as is phosphorylated H3.

R#3 further comment: I believe R#2 wants protein data. Are there GFP traps or something similar for AurA and/or AurB that can help since no Ab is available?

Reviewer #1:

Comment: I have examined the revised manuscript by Yang et al. Although the study is interesting and definitely worth pursuing, it is still too inconclusive to deserve publication in my opinion. Although the authors have satisfactorily addressed some of my concerns, at least two of my essential major concerns remain:

Response: We appreciate the reviewer's continued assessment of our work, and we are encouraged that a majority of the previous concerns were addressed in our revised manuscript.

Comment: The authors have not shown that Bsd directly phosphorylates Polo at T182. I understand that they are being careful with words to avoid drawing this conclusion explicitly, but this is still what they are leading readers to believe. The difficulty in expressing and purifying recombinant Polo from bacteria that they mention as an excuse is not insurmountable. For example, Carmena et al (2012, PLoS Biology, PMID 22291575) have expressed and purified HIS-Polo WT and T182A to show that Aurora B directly phosphorylates Polo in vitro (see Figure 4A in that paper). Until this crucial point is resolved, it will remain quite possible that Polo is still phosphorylated by Aurora B or Aurora A (or another kinase for that matter) in the context of muscle development studied here. The kinase activating Polo could simply depend on the activity of Bsd, either directly or indirectly.

Response: The reviewer's points are all correct. After some more optimization we were able to identify conditions to generate recombinant Polo protein from *E. coli*, and we found that Polo is a direct substrate of Bsd (**Fig S4E**).

Comment: The molecular mechanisms controlled by Polo in muscle morphogenesis need to be much better characterized. The direct substrate(s) of Polo in this process should be identified, and how its activity is modified by this regulation should be understood at a minimal level. These elements of molecular mechanisms directly upstream and downstream of Polo in muscle morphogenesis would need to be established for the paper to put forward reliable conclusions and constitute significant advance, in my opinion.

Response: The objectives outlined here are indeed noteworthy goals. We showed that the direct Polo binding protein Tum and Tum's direct binding partner Pav are essential for muscle morphogenesis. Our genetic analyses established Polo, Tum, and Pav act in a common pathway (**Figs 6A,B S6A-C**). Purifying Tum and Pav proteins to show direct phosphorylation, followed by in vitro experiments characterizing Tum and Pav localization in the presence of Polo and Bsd would indeed be exciting. But these experiments fall beyond the scope of an initial manuscript aimed at defining the function of a previously unknown kinase during myogenesis. We have included a further discussion on this topic, and elaborate on the idea that the Polo/Tum/Pav cytoskeletal regulatory module functions outside the mesoderm to regulate cellular guidance.

Reviewer #3:

Comment: The manuscript by Yang, et al., is largely improved and addresses major reviewer comments.

Response: Thank you for the continued suggestions to improve our manuscript.

Comment: Minor comments that should be corrected are 1) p. 4: The correct elongation and attachment of individual muscle results in a stereotypical morphology for each muscle that forms a largely invariant pattern in abdominal segments A2-A8 (Fig. 1B,C) should refer to Fig. 1A and 1B (not C).

Response: Yikes!! We have doubled checked ALL figure references for accuracy.

Comment: 2) Fig. 2A: no preimmune blot is shown as indicated in the figure legend.

Response: Another nice catch! We have double-checked all figure legends for accuracy.

Comment: 3) MW markers should be added to Western blots. Especially Fig. S2B, which states 'The Bsd and Myc antibodies recognized proteins of the same molecular weight.' While the myc tag is not large, you may be able to see a slight shift. Or rephrase this sentence.

Response: Thanks for the suggestion. Molecular weight markers were added to all Western Blots where possible. We have provided full size blots with molecular markers in the Source Data file.

Comment: 4) Why is 24B-Gal4 used for rescue while qPCR results for mef2-Gal4 are used (Fig 2)? These are not comparing the same things.

Response: Indeed they are not the same thing. We repeated the qPCR using 24B.Gal4. The results confirm the transgenes are expressed at equivalent levels.

Comment: 5) Please be consistent with I129 vs Iso129.

Response: Done!!

Comment: 6) p. 6: The manuscript states that 'Bsd did not promote the phosphorylation of Polo serine residues,' but this is not shown.

Response: Fair point. This sentence doesn't add much to the manuscript, so we deleted it as shown.

~~Bsd did not promote the phosphorylation of Polo serine residues, arguing Bsd specifically promotes the phosphorylation of Polo at threonine residues.~~

Comment: 7) Did pav enhance tum or polo mutant phenotypes? The experiment was only performed with polo, tum mutants.

Response: Thanks for the suggestion. *tum*^{DH15} did not enhance the *pav*^{B200} mutant

phenotype (**Fig. S6C**), further suggesting Polo, Tum, and Pav act in a common myogenic pathway.

Comment: 8) Please add embryonic stages to the embryo panels that are present in supplemental figures as you did for the main figures. Easier to see than in the figure legends as there are many different stages.

Response: Done!!

Comment: The 9) There is no source information for nod-GFP or the anti-gamma tubulin Ab. Please add.

Response: We included source information for these specific reagents and also include a new Supplemental Table 3 that includes source information for all reagents used in this study.

Comment: (From 1st review): Does VRK3 rescue the *bsd* phenotype?

Response: Yes!!!! We have included the human rescue in this revision (Fig. 7G).

Reviewer #3's further comments on responses to Reviewer #2's previous concerns

Comment: One important point is how clear is the expression data to support the statement in the Introduction (I57): “the transition from mitotic growth to cellular morphogenesis is achieved through the spatially and temporally restricted expression of the Aurora kinases and Bsd”.

Response: By Stage 12, aurA and aurB mRNA expression is restricted to the CNS as is phosphorylated H3.

R#3 further comment: It is not obvious from the pictures presented in Figure 5A that AurA and AurB expression becomes restricted to the CNS. This reviewer cannot really see much expression in the CNS and hence this comment should be amended.

Response: Thank you for the extra effort to complete these reviewers. We greatly appreciate it!!! With regards to this comment, you are correct that CNS expression was an assumption. We have removed CNS from our description, and state only “excluded from somatic mesoderm”. We have also included high magnification micrographs to show expression is restricted to the ventral and anterior regions of the embryo outside the somatic mesoderm (Fig. S5B).

Comment: Polo. It would be helpful to include more information on the polo hypomorph allele. Does it have a phenotype prior to muscle cell guidance? One would expect many polo mutants to have earlier phenotypes that might influence how muscle develops, e.g. affect an aspect of cell division that could affect the myoblast pool available for muscle development. As for bsd, the question also arises as to whether polo affects muscle precursor/founder specification. A gap in the manuscript is that only a stage 16 (near the end of embryogenesis) muscle phenotype is shown.

Response: Thank you for the suggestions. We had been operating under the assumption that maternally contributed polo allowed polo1 embryos to develop normally until myotube guidance initiates. In response to this comment, we assayed founder cell specification in polo1 embryos and found 5053>GFP and slou>GFP expressing founders were correctly specified (Fig. S3E,F). To clarify how the polo1 allele affects early embryonic development, we have added the following text: The polo1 allele is a maternal effect mutation that disrupts mitotic chromosome alignment [46], but zygotic mutants are partially viable and do not show mitotic defects until after embryogenesis [47]. (Lines 148-150).

R#3 further comment: This clarification does not address the point of R#2. If maternal load precludes analysis of phenotypes during most of embryogenesis, how can the authors conclude that founder cell specification occurs normally (which happens in mid embryogenesis). Can you make maternal clones with a weak allele of polo? Can you demonstrate the indeed maternal load is present during stages 12-16?

Response: Thank you for pointing out our explanation was unclear. We revised our explanation of maternal contribution, which is pasted at the end of this paragraph. In short, maternal contribution does not preclude analysis but rather allows *polo* zygotic mutant embryos to develop normally through critical times of mitosis. Previous studies identified mitotic phenotypes in embryos derived from maternal (germline) clones,

whereas embryos that were only zygotic mutant had no issues with mitosis (at least in the stages assayed). Thus, the maternal contribution of *polo* reduces mitotic phenotypes in zygotic mutants. Since germline clones would disrupt development prior to myogenesis, we instead assayed the total number of mesoderm cells during founder cell specification (**Fig. S3E**). There is a slight but significant decrease in the number of mesoderm cells in *polo* mutant embryos, but founder cell number was not significantly different from wild-type (**Fig. S3G-I**). These observations suggest that the mitotic defects in *polo*¹ zygotic mutant embryos do not appreciably affect the myoblast pool available for muscle development.

Germ line clones of the hypomorphic allele *polo*¹ reduce the maternal contribution of Polo, and *polo*¹ maternal mutant embryos showed defects in mitotic chromosome alignment [34]. However, *polo*¹ zygotic mutants with a normal maternal contribution of Polo were largely viable, and did not show embryonic mitotic defects [35]. After gastrulation, the mesoderm undergoes four rounds of cell division that conclude by Stage 11 [36]. During Stages 10-11, each founder cell is specified from a pool of cells known as an equivalence group. The remaining cells in the equivalence group then differentiate into fusion competent myoblasts [25]. To understand if *polo*¹ zygotic mutant embryos undergo normal cell divisions prior to founder cell specification, we quantified the number of Mef2-positive somatic mesoderm cells in Stage 10 embryos. Although *polo*¹ embryos showed a 14.2% decrease in the number Mef2-positive cells compared to wild-type embryos (Fig. S3E,F), the number of *5053.Gal4* and *slou.gal4* expressing founder cells were not significantly different between wild-type and *polo*¹ Stage 12 embryos (Fig. S3G,H). These observations suggest that the mitotic defects in *polo*¹ zygotic mutant embryos do not appreciably affect equivalence group size or founder cell specification. Using *5053.Gal4* and *slou.gal4* to characterize muscle morphogenesis, we identified myotube elongation and muscle attachment site selection defects in the VL1, DT1, L01, and VT1 muscles of *polo*¹ embryos (Fig. 3D,E), suggesting Polo directs myotube guidance after founder cell specification. Importantly, the severity of muscle morphogenetic defects was equivalent among *bsd*¹, *polo*¹, and *bsd*¹ *polo*¹ embryos, arguing Bsd and Polo act in a common pathway to direct myotube guidance (Figs. 3F-H, S3I).

Comment: What is the evidence that there is no mesodermal Aurora expression after this point? The authors just cite refs 44 and 45 in support. Both are in situ hybridisations (not protein) with little detail. 45 is the bulk assay of gene expression in the Drosophila genome project. These citations do not seem sufficient for this important point. This really needs the authors to include their own high quality study specifically addressing the question of the timing of any Aur B (and A) expression in the mesoderm.

Response: Great suggestion. As stated previously, we did fluorescent in situ hybridization for *aurA* and *aurB*. We detected high levels of each mRNA in the somatic mesoderm at Stage 10, which correlates with H3 phosphorylation in the somatic mesoderm (we made a new figure for all of the Aurora-related revisions, Fig. 5A shows the in situ). By Stage 12, *aurA* and *aurB* mRNA expression is restricted to the CNS as is phosphorylated H3.

R#3 further comment: I believe R#2 wants protein data. Are there GFP traps or something similar for AurA and/or AurB that can help since no Ab is available?

Response: We have requested the antibody multiple times (to no avail), and unfortunately there are no insertions in the correct genomic location and in the correct orientation to substitute for or to generate an endogenous tag. However, Reviewer 2 simply stated “the authors [need] to include their own high quality study specifically addressing the question of the timing of any Aur B (and A) expression in the mesoderm”. We did address this comment in detail by performing our own in situ hybridizations for both *aurA* and *aurB*. In that regard, we feel the essence of the concern has been attended to experimentally.

Reviewers' Comments:

Reviewer #3:

Remarks to the Author:

All concerns have been satisfied. The paper should be accepted and published.